# Going Beyond Linear RL: Sample Efficient Neural Function Approximation

**Baihe Huang[1]\*, Kaixuan Huang[2]\*, Sham M. Kakade[3,4]\*, Jason D. Lee[2]\***
**Qi Lei[2]\*, Runzhe Wang[2]\*, Jiaqi Yang[5]\***

[1]Peking University    [2]Princeton University    [3]Harvard University
[4]Microsoft Research    [5]Tsinghua University

## Abstract

Deep Reinforcement Learning (RL) powered by neural net approximation of the Q function has had enormous empirical success. While the theory of RL has traditionally focused on linear function approximation (or eluder dimension) approaches, little is known about nonlinear RL with neural net approximations of the Q functions. This is the focus of this work, where we study function approximation with two-layer neural networks (considering both ReLU and polynomial activation functions). Our first result is a computationally and statistically efficient algorithm in the generative model setting under completeness for two-layer neural networks. Our second result considers this setting but under only realizability of the neural net function class. Here, assuming deterministic dynamics, the sample complexity scales linearly in the algebraic dimension. In all cases, our results significantly improve upon what can be attained with linear (or eluder dimension) methods.

## 1 Introduction

In reinforcement learning (RL), an agent aims to learn the optimal decision-making rule by interacting with an unknown environment [71]. Deep Reinforcement Learning, empowered by deep neural networks [48, 32], has achieved tremendous success in various real-world applications, such as Go [68], Atari [54], Dota2 [7], Texas Holdém poker [56], and autonomous driving [66]. Those modern RL applications are characterized by large state-action spaces, and the empirical success of deep RL corroborates the observation that deep neural networks can extrapolate well across state-action spaces [35, 55, 50].

Although in practice non-linear function approximation scheme is prevalent, theoretical understandings of the sample complexity of RL mainly focus on tabular or linear function approximation settings [69, 38, 5, 41, 63, 88, 1, 43, 44, 77]. These results rely on finite state space or exact linear approximations. Recently, sample efficient algorithms under non-linear function approximation settings are proposed [82, 11, 20, 12, 51, 72, 13, 91, 86]. Those algorithms are based on Bellman rank [40], eluder dimension [64], neural tangent kernel [37, 4, 14, 92], or sequential Rademacher complexity [60, 61]. Yet, the understanding of how deep RL learns and generalizes in large state spaces is far from complete. While the aforementioned works study function approximation structures that possess the nice properties of linear models, such as low information gain and low eluder dimensions, the highly non-linear nature of neural networks renders challenges on their applicability to deep RL. For one thing, recent wisdoms in deep learning theory cast doubt on the ability of neural tangent kernel and random features to model the actual neural networks. Indeed, the neural tangent kernel

---

\*Alphabetical order. Correspondence to: Baihe Huang, `baihehuang@pku.edu.cn`, Jason D. Lee, `Jasondl@princeton.edu`.

approximately reduces neural networks to linear models, but the RKHS norm of neural networks is exponential [87]. Moreover, it remains unclear what neural networks models have low eluder dimensions. For example, recent work [13] shows that two layer neural networks have exponential eluder dimension in the dimension of features. Thus, the mismatch between the empirical success of deep RL and its theoretical understanding remains significant, which yields the following important question:

*What are the structural properties that allow sample-efficient algorithms for RL with neural network function approximation?*

Recent work in RL suggests that learning RL with neural networks function approximation is exponentially hard [13, 49, 52]. In this paper, however, we advance the understanding of the above question by displaying several structural properties that allow efficient RL algorithms with neural function approximations. We consider several value function approximation models that possess high information gain and high eluder dimension. Specifically, we study two structures, namely two-layer neural networks and structured polynomials (i.e. two-layer neural networks with polynomial activation functions), under two RL settings, namely RL with simulator model and online RL. In the simulator (generative model) setting [45, 67], the agent can simulate the MDP at any state-action pair. In online RL, the agent can only start at an initial state and interact with the MDP step by step. The goal in both settings is to find a near-optimal policy while minimizing the number of samples used.

We obtain the following results. For the simulator setting, we propose sample-efficient algorithms for RL with two-layer neural network function approximation. Under either policy completeness, Bellman completeness, or gap conditions, our method provably learns near-optimal policy with polynomial sample complexities. For online RL, we provide sample-efficient algorithms for RL with structured polynomial function approximation. When the transition is deterministic, we also present sample-efficient algorithms under only the realizability assumption [18, 78]. Our main techniques are based on neural network recovery [90, 39, 28], and algebraic geometry [53, 65, 8, 76].

## 1.1 Summary of our results

Our main results in different settings are summarized in Table 1. We consider two-layer neural networks $f(x) = \langle v, \sigma(Wx) \rangle$ (where $\sigma$ is ReLU activation) and rank $k$ polynomials (see Example 4.3).

Table 1: Baselines and our main results for the sample complexity to find an $\epsilon$-optimal policy.

|  | rank $k$ polynomial | | Neural Net of Width $k$ | | |
|---|---|---|---|---|---|
|  | Sim. + Det. (R) | Onl. + Det. (R) | Sim. + Det. (R) | Sim. + Gap. (R) | Sim. + Stoch. (C) |
| Baseline | $O(d^p)$ | $O(d^p)$ | $O(d^{\mathrm{poly}(1/\epsilon)})$ (*) | $O(d^{\mathrm{poly}(1/\epsilon)})$ | $O(d^{\mathrm{poly}(1/\epsilon)})$ |
| Our results | $O(dk)$ | $O(dk)$ | $\widetilde{O}(\mathrm{poly}(d) \cdot \exp(k))$ | $\widetilde{O}(\mathrm{poly}(d,k))$ | $\widetilde{O}(\mathrm{poly}(d,k)/\epsilon^2)$ |

We make the following elaborations on Table 1.

- For simplicity, we display only the dependence on the feature dimension $d$, network width or polynomial rank $k$, precision $\epsilon$, and degree $p$ (of polynomials).

- In the table *Sim.* denotes simulator model, *Onl.* denotes online RL, *Det.* denotes deterministic transitions, *Stoch.* denotes stochastic transitions, *Gap.* denotes gap condition, *(R)* denotes realizability assumption only, and *(C)* denotes completeness assumption (either policy complete or Bellman complete) together with realizability assumption.

- We apply [21] for the deterministic transition baseline, and apply [19] for the stochastic transition baseline. We are unaware of any methods that can directly learn MDP with neural network value function approximation[2].

- In polynomial case, the baseline first vectorizes the tensor $\begin{pmatrix} 1 \\ x \end{pmatrix}^{\otimes p}$ into a vector in $\mathbb{R}^{(d+1)^p}$ and then performs on this vector. In the neural network case, the baseline uses a polynomial of degree $1/\epsilon$ to approximate the neural network with precision $\epsilon$ and then vectorizes the polynomial into a vector in $\mathbb{R}^{d^{\mathrm{poly}(1/\epsilon)}}$. The baseline method for realizable model (denoted

by (*)) needs a further gap assumption of gap $\geq d^{\text{poly}(1/\epsilon)}\epsilon$ to avoid the approximation error from escalating [21]; note for small $\epsilon$ this condition never holds but we include it in the table for the sake of comparison.

- In rank $k$ polynomial case, our result $O(dk)$ in simulator model can be found in Theorem 4.7 and our result $O(dk)$ in online RL model can be found in Theorem 4.8. These results only require a realizability assumption. Efficient explorations are guaranteed by algebraic-geometric arguments. In neural network model, our result $\widetilde{O}(\text{poly}(d)\cdot\exp(k))$ in simulator model can be found in Theorem 3.4. This result also only relies on the realizability assumption. For stochastic transitions, our result $\widetilde{O}(\text{poly}(d,k)/\epsilon^2)$ works for either policy complete or Bellman complete settings, as in Theorem 3.5 and Theorem 3.6 respectively. The $\widetilde{O}(\text{poly}(d,k))$ result for gap condition can be found in Theorem 3.8.

## 1.2 Related Work

**Linear Function Approximation.** RL with linear function approximation has been widely studied under various settings, including linear MDP and linear mixture MDP [44, 89, 85]. While these papers have proved efficient regret and sample complexity bounds, their analyses relied heavily on two techniques: they used the confidence ellipsoid to quantify the uncertainty, and they used the elliptical potential lemma to bound the total uncertainty [2]. These two techniques were integral to their analyses but are so restrictive that they generally do not extend to nonlinear cases.

**Eluder Dimension.** [62, 59] proposed eluder dimension, a complexity measure of the function space, and proved regret and sample complexity bounds that scaled with the eluder dimension, for bandits and reinforcement learning [73, 42]. They also showed that the eluder dimension is small in several settings, including generalized linear models and LQR. However, as shown in [13], the eluder dimension could be exponentially large even with a single ReLU neuron, which suggested the eluder dimension would face difficulty in dealing with neural network cases. The eluder dimension is only known to give non-trivial bounds for linear function classes and monotone functions of linear function classes. For structured polynomial classes, the eluder dimension simply embeds into an ambient linear space of dimension $d^p$, where $d$ is the dimension, and $p$ is the degree. This parallels the lower bounds in linearization / neural tangent kernel (NTK) works [79, 30, 3], which show that linearization also incurs a similarly large penalty of $d^p$ sample complexity, and more advanced algorithm design is need to circumvent linearization[6, 9, 22, 83, 26, 58, 29, 57, 34, 75, 10].

**Bellman Rank and Completeness.** [40, 70] studied RL with general function approximation. They used Bellman rank to measure the error of the function class under the Bellman operator and gave proved bounds in the term of it. Recently, [16] propose bilinear rank and encompass more function approximation models. However, it is hard to bound either the Bellman rank or the bilinear rank for neural nets. Therefore, their results are not known to include the neural network approximation setting. Another line of work shows that exponential sample complexity is unavoidable even with good representations [19, 80, 78], which implies the realizability assumption alone might be insufficient for function approximations.

## 2 Preliminaries

An episodic Markov Decision Process (MDP) is defined by the tuple $\mathcal{M} = (\mathcal{S}, \mathcal{A}, H, \mathbb{P}, r)$ where $\mathcal{S}$ is the state space, $\mathcal{A}$ is the action set, $H$ is the number of time steps in each episode, $\mathbb{P}$ is the transition kernel and $r$ is the reward function. In each episode the agent starts at a fixed initial state $s_1$ and at each time step $h \in [H]$ it takes action $a_h$, receives reward $r_h(s_h, a_h)$ and transits to $s_{h+1} \sim \mathbb{P}(\cdot|s_h, a_h)$.

A deterministic policy $\pi$ is a length-$H$ sequence of functions $\pi = \{\pi_h : \mathcal{S} \mapsto \mathcal{A}\}_{h=1}^{H}$. Given a policy $\pi$, we define the value function $V_h^\pi(s)$ as the expected sum of reward under policy $\pi$ starting from

---

[2] Prior work on neural function approximation has focused on neural tangent kernels, which would require $d^{\text{poly}(1/\epsilon)}$ to approximate a two-layer network [31].

$s_h = s$:

$$V_h^\pi(s) := \mathbb{E}\left[\sum_{t=h}^H r_t(s_t, a_t)|s_h = s\right]$$

and we define the Q function $Q_h^\pi(s, a)$ as the the expected sum of reward taking action $a$ in state $s_h = s$ and then following $\pi$:

$$Q_h^\pi(s, a) := \mathbb{E}\left[\sum_{t=h}^H r_t(s_t, a_t)|s_h = s, a_h = a\right].$$

The Bellman operator $\mathcal{T}_h$ applied to Q-function $Q_{h+1}$ is defined as follow

$$\mathcal{T}_h(Q_{h+1})(s, a) := r_h(s, a) + \mathbb{E}_{s' \sim \mathbb{P}(\cdot|s,a)}[\max_{a'} Q_{h+1}(s', a')].$$

There exists an optimal policy $\pi^*$ that gives the optimal value function for all states, i.e. $V_h^{\pi^*}(s) = \sup_\pi V_h^\pi(s)$ for all $h \in [H]$ and $s \in \mathcal{S}$. For notational simplicity we abbreviate $V^{\pi^*}$ as $V^*$ and correspondingly $Q^{\pi^*}$ as $Q^*$. Therefore $Q^*$ satisfies the following Bellman optimality equations for all $s \in \mathcal{S}$, $a \in \mathcal{A}$ and $h \in [H]$:

$$Q_h^*(s, a) = \mathcal{T}_h(Q_{h+1}^*)(s, a).$$

The goal is to find a policy $\pi$ that is $\epsilon$-optimal in the sense that $V_1^*(s_1) - V_1^\pi(s_1) \leq \epsilon$, within a small number of samples. We consider two query models of interacting with MDP:

- In the simulator model ([45], [67]), the agent interacts with a black-box that simulates the MDP. At each time step $h \in [H]$, the agent can start at a state-action pair $(s, a)$ and interact with the black box by executing some policy $\pi$ chosen by the agent.

- In online RL, the agent can only start at the initial state and interact with the MDP by using a policy and observing the rewards and the next states. In each episode $k$, the agent proposes a policy $\pi^k$ based on all history up to episode $k - 1$ and executes $\pi^k$ to generate a single trajectory $\{s_h^k, a_h^k\}_{h=1}^H$ with $a_h^k = \pi_h^k(s_h^k)$ and $s_{h+1}^k \sim \mathbb{P}(\cdot|s_h^k, a_h^k)$.

## 2.1 Function approximation

In reinforcement learning with value function approximation, the learner is given a function class $\mathcal{F} = \mathcal{F}_1 \times \cdots \times \mathcal{F}_H$, where $\mathcal{F}_h \subset \{f : \mathcal{S} \times \mathcal{A} \mapsto [0, 1]\}$ is a set of candidate functions to approximate $Q^*$. The following assumption is commonly adopted in the literature [43, 74, 42, 17].

**Assumption 2.1** (Realizability). $Q_h^* \in \mathcal{F}_h$ for all $h \in [H]$.

The function approximation is equipped with feature mapping $\phi : \mathcal{S} \times \mathcal{A} \mapsto \{u \in \mathbb{R}^d : \|u\|_2 \leq B_\phi\}$ that is known to the agent. We focus the continuous action setting (e.g. in control and robotics problems) and make the following regularity assumption on the feature function $\phi$.

**Assumption 2.2** (Bounded features). Assume $\phi(s, a) \leq B_\phi, \forall(s, a) \in \mathcal{S} \times \mathcal{A}$.

**Notation** For any vector $x \in \mathbb{R}^d$, let $x_{\max} := \max_{i \in [d]} x_i$ and $x_{\min} := \min_{i \in [d]} x_i$. Let $s_i(\cdot)$ denote the $i$-th singular value, $s_{\min}(\cdot)$ denotes the minimum eigenvalue and $s_{\max}(\cdot)$ denotes the maximum eigenvalue. The conditional number is defined by $\kappa(\cdot) := s_{\max}(\cdot)/s_{\min}(\cdot)$. We use $\otimes$ to denote Kronecker product and $\circ$ to denote Hadamard product. For a given integer $H$, we use $[H]$ to denote the set $\{1, 2, \ldots, H\}$. For a function $f : \mathfrak{X} \mapsto \mathfrak{Y}$, we use $f^{-1}(y) := \{x \in \mathfrak{X} : f(x) = y\}$ to denote the preimage of $y \in \mathfrak{Y}$. We use the shorthand $x \lesssim y$ ($x \gtrsim y$) to indicate $x \leq O(y)$ ($x \geq \Omega(y)$).

# 3 Neural Network Function Approximation

In this section we show sample-efficient algorithms with neural network function approximations. The function class of interest is given in the following definition. More general neural network class is discussed in Appendix B.5.

**Definition 3.1** (Neural Network Function Class). We use $\mathcal{F}_{NN}$ to denote the function class of $f(\phi(s,a)): \mathcal{S} \times \mathcal{A} \mapsto \mathbb{R}$ where $f(x) = \langle v, \sigma(Wx) \rangle : \mathbb{R}^d \mapsto \mathbb{R}$ is a two-layer neural network where $\sigma$ is ReLU, $\|W\|_F \leq B_W$, $v \in \{\pm 1\}^k$, $\prod_{i=1}^k s_i(W)/s_{\min}(W) \leq \lambda$, $s_{\max}(W)/s_{\min}(W) \leq \kappa$ and $k \leq d$. Here $\phi : \mathcal{A} \times \mathcal{S} \mapsto \mathbb{R}^d$ is a known feature map whose image contains a ball $\{u \in \mathbb{R}^d : \|u\|_2 \leq \delta_\phi\}$ with $\delta_\phi \geq d \cdot \mathrm{polylog}(d)$.[3]

We introduce the following completeness properties in the setting of value function approximations. Along with Assumption 2.1, they are commonly adopted in the literature .

**Definition 3.2** (Policy complete). Given MDP $\mathcal{M} = (\mathcal{S}, \mathcal{A}, \mathbb{P}, r, H)$, function class $\mathcal{F}_h : \mathcal{S} \times \mathcal{A} \mapsto \mathbb{R}, h \in [H]$ is called policy complete if for all $\pi$ and $h \in [H]$, $Q_h^\pi \in \mathcal{F}_h$.

**Definition 3.3** (Bellman complete). Given MDP $\mathcal{M} = (\mathcal{S}, \mathcal{A}, \mathbb{P}, r, H)$, function class $\mathcal{F}_h : \mathcal{S} \times \mathcal{A} \mapsto \mathbb{R}, h \in [H]$ is called Bellman complete if for all $h \in [H]$ and $Q_{h+1} \in \mathcal{F}_{h+1}$, $\mathcal{T}_h(Q_{h+1}) \in \mathcal{F}_h$.

### 3.1 Warmup: Realizable $Q^*$ with deterministic transition

We start by considering the case when the transition kernel is deterministic. In this case only Assumption 2.1 is required for the expressiveness of neural network function approximations. Algorithm 1 learns optimal policy from time step $H$ to 1. Suppose we have learned policies $\pi_{h+1}, \dots, \pi_H$ at level $h$ and they are exactly the optimal policies. We first explore features $\phi(s_h^i, a_h^i)$ over a standard Gaussian distribution, and if $\|\phi(s_h^i, a_h^i)\|_2 \geq \delta_\phi$ then we simply skip this trial. Recall that $\delta_\phi \geq d \cdot \mathrm{poly}\log(d)$, so with high probability (w.r.t $d$) almost all feature samples will be explored. We next construct an estimate $\widehat{Q}_h^i$ of $Q^*(s_h^i, a_h^i)$ by collecting cumulative rewards using $\pi_{h+1}, \dots, \pi_H$ as the roll-out. Since the transition is deterministic, $\widehat{Q}_h^i = Q^*(s_h^i, a_h^i)$ for all explored samples $(s_h^i, a_h^i)$. Recall that $Q_h^*$ is a two-layer neural network, we can now recover its parameters in Line 12 exactly by invoking techniques in neural network optimization (see, e.g. [39, 27, 90]). Details of this step can be found in Appendix B.5, where the method is mainly based on [90]. This means the reconstructed $\widehat{Q}_h$ in Line 13 is precisely $Q^*$, and the algorithm can thus find optimal policy $\pi_h^*$ in the $h$-th level.

---

**Algorithm 1** Learning realizable $Q^*$ with deterministic transition

1: **for** $h = H, \dots 1$ **do**
2:      Sample $x_h^i, i \in [n]$ from standard Gaussian $\mathcal{N}(0, I_d)$
3:      **for** $i \in [n]$ **do**
4:          **if** $\|x_h^i\| \leq \delta_\phi$ **then**
5:              Find $(s_h^i, a_h^i) \in \phi^{-1}(x_h^i)$ and locate the state $s_h^i$ in the generative model
6:              Pull action $a_h^i$ and use $\pi_{h+1}, \dots, \pi_H$ as the roll-out to collect rewards $r_h^{(i)}, \dots, r_H^{(i)}$
7:              Construct estimation

$$\widehat{Q}_h^i \leftarrow r_h^{(i)} + \cdots + r_H^{(i)}$$

8:          **else**
9:              Let $\widehat{Q}_h^i \leftarrow 0$.
10:          **end if**
11:      **end for**
12:      Compute $(v_h, W_h) \leftarrow \textsc{NeuralNetRecovery}(\{(x_h^i, \widehat{Q}_h^i) : i \in [n]\})$
13:      Set $\widehat{Q}_h(s,a) \leftarrow v_h^\top \sigma(W_h \phi(s,a))$
14:      Let $\pi_h(s) \leftarrow \arg\max_{a \in \mathcal{S}} \widehat{Q}_h(s,a)$
15: **end for**
16: **Return** $\pi_1, \dots, \pi_H$

---

**Theorem 3.4.** *(Informal) If $n \geq d \cdot \mathrm{poly}(\kappa, k, \lambda, \log d, B_W, B_\phi, H)$, then with high probability Algorithm 1 learns the optimal policy.*

The formal statement and complete proof are deferred to the Appendix B.1. The main idea of exact neural network recovery can be summarized in the following. We first use method of moments to

---

[3]Here the $\delta_\phi$ is chosen only for simplicity. In general this assumption can be relaxed to that the image of $\phi$ contains an arbitrary dense ball near the origin, since one can always rescale the feature mapping in the neural function approximation.

find a 'rough' parameter recovery. If this 'rough' recovery is sufficiently close to the true parameter, the empirical $\ell_2$ loss function is locally strongly convex and there is unique global minimum. Then we can apply gradient descent to find this global minimum which is exactly the true parameter.

## 3.2 Policy complete neural function approximation

Now we consider general stochastic transitions. Difficulties arise in this scenario due to noises in the estimation of Q-functions. In the presence of model misspecification, these noises cause estimation errors to amplify through levels and require samples to be exponential in $H$. In this section, we show that neural network function approximation is still learnable, assuming the function class $\mathcal{F}_{NN}$ is policy complete with regard to MDP $\mathcal{M}$. Thus for all $\pi \in \Pi$, we can denote $Q_h^\pi(s, a) = \langle v^\pi, \sigma(W^\pi \phi(s, a)) \rangle$.

---

**Algorithm 2** Learn policy complete NN with simulator.

---

1: **for** $h = H, \ldots 1$ **do**
2:     Sample $x_h^i, i \in [n]$ from standard Gaussian $\mathcal{N}(0, I_d)$
3:     **for** $i \in [n]$ **do**
4:         **if** $\|x_h^i\| \leq \delta_\phi$ **then**
5:             Find $(s_h^i, a_h^i) \in \phi^{-1}(x_h^i)$ and locate the state $s_h^i$ in the generative model
6:             Pull action $a_h^i$ and use $\pi_{h+1}, \ldots, \pi_H$ as the roll-out to collect rewards $r_h^{(i)}, \ldots, r_H^{(i)}$
7:             Construct unbiased estimation of $Q_h^{\pi_{h+1}, \ldots, \pi_H}(s_h^i, a_h^i)$

$$\widehat{Q}_h^i \leftarrow r_h^{(i)} + \cdots + r_H^{(i)}$$

8:         **else**
9:             Let $\widehat{Q}_h^i \leftarrow 0$.
10:         **end if**
11:     **end for**
12:     Compute $(v_h, W_h) \leftarrow \text{NEURALNETNOISYRECOVERY}(\{(x_h^i, \widehat{Q}_h^i) : i \in [n]\})$
13:     Set $\widehat{Q}_h(s, a) \leftarrow v_h^\top \sigma(W_h \phi(s, a))$
14:     Let $\pi_h(s) \leftarrow \arg\max_{a \in \mathcal{S}} \widehat{Q}_h(s, a)$
15: **end for**
16: **Return** $\pi_1, \ldots, \pi_H$

---

Algorithm 2 learns policy from level $H, H-1, \ldots, 1$. In level $h$, the algorithm has learned policy $\pi_{h+1}, \ldots, \pi_H$ that is only sub-optimal by $(H - h)\epsilon/H$. Then it explores features $\phi(s, a)$ from $\mathcal{N}(0, I_d)$. The algorithm then queries $(s, a)$ and uses learned policy $\pi_{h+1}, \ldots, \pi_H$ as roll out to collect an unbiased estimate of the Q-function $Q_h^{\pi_{h+1}, \ldots, \pi_H}(s, a)$. Since $Q_h^{\pi_{h+1}, \ldots, \pi_H}(s, a) \in \mathcal{F}_{NN}$ is a two-layer neural network, it can then be recovered from samples. Details of this step can be found in Appendix B.5, where the methods are mainly based on [90]. The algorithm then reconstructs this Q-function and finds its optimal policy $\pi_h$.

**Theorem 3.5.** *(Informal) Fix $\epsilon, t$, if $n \geq \epsilon^{-2} \cdot d \cdot \text{poly}(\kappa, k, B_W, B_\phi, H, \log(d/t))$, then with probability at least $1 - t$ Algorithm 2 returns an $\epsilon$-optimal policy $\pi$.*

The formal statement and complete proof are deferred to Appendix B.2. Notice that unlike the case of Theorem 3.4, the sample complexity does not depend on $\lambda$, thus avoiding the potential exponential dependence in $k$.

The main idea of the proof is that at each time step a neural network surrogate of $Q^*$ can be constructed by the policy already learned. Suppose we have learned $\pi_{h+1}, \ldots, \pi_H$ in level $h$, then from policy completeness $Q_h^{\pi_{h+1}, \ldots, \pi_H}$ belongs to $\mathcal{F}_{NN}$ and we can interact with the simulator to obtain its estimate $\widehat{Q}_h$. If $\|\widehat{Q}_h - Q_h^{\pi_{h+1}, \ldots, \pi_H}\|_\infty$ is small, the optimistic planning based on $\widehat{Q}_h$ is not far from the optimal policy of $Q_h^{\pi_{h+1}, \ldots, \pi_H}$. Therefore the errors can be decoupled into the errors in recovering $Q_h^{\pi_{h+1}, \ldots, \pi_H}$ and the suboptimality of $Q_h^{\pi_{h+1}, \ldots, \pi_H}$, which depends on level $h + 1$. This reasoning can then be recursively performed to level $H$, and thus we can bound the suboptimality of $\pi_h$.

### 3.3 Bellman complete neural function approximation

In addition to policy completeness, we show that neural network function approximation can also learn efficiently under the setting where the function class $\mathcal{F}_{NN}$ is Bellman complete with regard to MDP $\mathcal{M}$. Specifically, for $Q_{h+1} \in \mathcal{F}_{h+1}$, there are $v^{Q_{h+1}}$ and $W^{Q_{h+1}}$ such that $\mathcal{T}_h(Q_{h+1})(s,a) = \langle v^{Q_{h+1}}, \sigma(W^{Q_{h+1}}\phi(s,a)) \rangle$.

Algorithm 3 is similar to the algorithm in previous section. Suppose in level $h$, the algorithm has constructed the Q-function $\widehat{Q}_{h+1}(s,a) = v_{h+1}^\top \sigma(W_{h+1}\phi(s,a))$ that is $(H-h)\epsilon/H$-close to the optimal $Q_{h+1}^*$. It then recovers weights $v_h, W_h$ from $\mathcal{T}_h(\widehat{Q}_{h+1})(s,a) = \langle v^{\widehat{Q}_{h+1}}, \sigma(W^{\widehat{Q}_{h+1}}\phi(s,a)) \rangle$, using unbiased estimates $r_h(s_h^i, a_h^i) + \widehat{V}_{h+1}(s_{h+1}^i)$. The Q-function $\widehat{Q}_h(s,a) = v_h^\top \sigma(W_h\phi(s,a))$ reconstructed from weights $v_h, W_h$ is thus $(H-h+1)\epsilon/H$-close to the $Q_h^*$.

---

**Algorithm 3** Learn Bellman complete NN with simulator.

---

1: **for** $h = H, \ldots 1$ **do**
2:     Sample $x_h^i, i \in [n]$ from standard Gaussian $\mathcal{N}(0, I_d)$
3:     **for** $i \in [n]$ **do**
4:         **if** $\|x_h^i\| \le \delta_\phi$ **then**
5:             Find $(s_h^i, a_h^i) \in \phi^{-1}(x_h^i)$ and locate the state $s_h^i$ in the generative model
6:             Pull action $a_h^i$ and and observe $r_h(s_h^i, a_h^i), s_{h+1}^i$
7:             Construct unbiased estimation of $\mathcal{T}_h(\widehat{Q}_{h+1})(s_h^i, a_h^i)$

$$\widehat{Q}_h^i \leftarrow r_h(s_h^i, a_h^i) + \widehat{V}_{h+1}(s_{h+1}^i)$$

8:         **else**
9:             Let $\widehat{Q}_h^i \leftarrow 0$.
10:         **end if**
11:     **end for**
12:     Compute $(v_h, W_h) \leftarrow \text{NEURALNETNOISYRECOVERY}(\{(x_h^i, \widehat{Q}_h^i) : i \in [n]\})$
13:     Set $\widehat{Q}_h(s,a) \leftarrow v_h^\top \sigma(W_h\phi(s,a))$ and $\widehat{V}_h \leftarrow \max_{a \in \mathcal{A}} \widehat{Q}_h(s,a)$
14:     Let $\pi_h(s) \leftarrow \arg\max_{a \in \mathcal{A}} \widehat{Q}_h(s,a)$
15: **end for**
16: **Return** $\pi_1, \ldots, \pi_H$

---

**Theorem 3.6.** *(Informal) Fix $\epsilon, t$, if $n \ge \epsilon^{-2} \cdot d \cdot \text{poly}(\kappa, k, B_W, B_\phi, H, \log(d/t))$, then with probability at least $1 - t$ Algorithm 3 returns an $\epsilon$-optimal policy $\pi$.*

Due to Bellman completeness, the error of estimation $\widehat{Q}_h$ can be controlled recursively. In fact, we can show $\|\widehat{Q}_h - Q^*(s,a)\|_\infty$ is small by induction. The formal statement and detailed proof are deferred to Appendix B.3. Similar to Theorem 3.5, the sample complexity does not explicitly depend on $\lambda$, thus avoiding potentially exponential dependence in $k$.

### 3.4 Realizable $Q^*$ with optimality gap

In this section we consider MDPs where there is a non-zero gap between the optimal policy and any other ones. This concept, known as optimality gap, is widely used in reinforcement learning and bandit literature [20, 19, 21].

**Definition 3.7.** The optimality gap is defined as

$$\rho = \inf_{a:Q^*(s,a) \neq V^*(s)} V^*(s) - Q^*(s,a).$$

We show that in the presence positive optimality gap, there exists an algorithm that can learn the optimal policy with polynomial samples even without the completeness assumptions. Intuitively, this is because one only needs to recover the neural network up to precision $\rho/4$ in order to make sure the greedy policy is identical to the optimal one. The formal statement and proof are deferred to Appendix B.4.

**Theorem 3.8.** *(Informal) Fix $t \in (0, 1)$, if $n = \frac{d}{\rho^2} \cdot \mathrm{poly}(\kappa, k, B_W, B_\phi, H, \log(d/t))$, then with probability at least $1 - t$ there exists an algorithm that returns the optimal policy $\pi^*$.*

**Remark 3.9.** *In all aforementioned methods, there are two key components that allow efficient learning. First, the exploration is conducted in a way that guarantees an $\ell_\infty$ recovery of candidate functions. By $\ell_\infty$ recovery we mean the algorithm recovers a candidate Q-function in this class deviating from the target function $Q^*$ by at most $\epsilon$ uniformly for all state-action pairs in the domain of interest. This notion of learning guarantee has received study in active learning [33, 46] and recently gain interest in contextual bandits [25]. Second, the agent constructs unbiased estimators of certain approximations to $Q^*$ that lie in the neural function approximation class. This allows the recovery error to decouple linearly across time steps, which is made possible in several well-posed MDP instances, such as deterministic MDPs, MDPs with completeness assumptions, and MDPs with gap conditions. In principle, we note that provably efficient RL algorithms with general function approximation is possible as long as the above two components are present. We will see in the next section another example of learning RL with highly non-convex function approximations, where the function class of interest, admissible polynomial families, also allows for exploration schemes to achieve $\ell_\infty$ recovery.*

## 4 Polynomial Realizability

In this section, we study the sample complexity to learn deterministic MDPs under polynomial realizability. We identify sufficient and necessary conditions for efficiently learning the MDPs for two different settings — the generative model setting and the online RL setting. Specifically, we show that if the image of the feature map $\phi_h(s_h, a_h)$ satisfies some positive measure conditions, then by random exploring, we can identify the optimal policy with samples linear in the algebraic dimension of the underlying polynomial class. We also provide a lower bound example showing the separation between the two settings.

Next, we introduce the notion of **Admissible Polynomial Families**, which are the families of structured polynomials that enable efficient learning.

**Definition 4.1** (Admissible Polynomial Families). For $x \in \mathbb{R}^d$, denote $\widetilde{x} = [1, x^\top]^\top$. Let $\mathcal{X} := \left\{ \widetilde{x}^{\otimes p} : x \in \mathbb{R}^d \right\}$. For any algebraic variety $\mathcal{V}$, we define $\mathcal{F}_\mathcal{V} := \{ f_\Theta(x) = \langle \Theta, \widetilde{x}^{\otimes p} \rangle : \Theta \in \mathcal{V} \}$ as the polynomial family parameterized by $\Theta \in \mathcal{V}$. We say $\mathcal{F}_\mathcal{V}$ is admissible[4] w.r.t. $\mathcal{X}$, if for any $\Theta \in \mathcal{V}$, $\dim(\mathcal{X} \cap \{ X \in \mathcal{X} : \langle X, \Theta \rangle = 0 \}) < \dim(\mathcal{X}) = d$. We define the dimension $D$ of the family to be the dimension of $\mathcal{V}$.

The following theorem shows that to learn an admissible polynomial family, the sample complexity only scales with the algebraic dimension of the polynomial family.

**Theorem 4.2** ([36]). *Consider the polynomial family $\mathcal{F}_\mathcal{V}$ of dimension $D$. For $n \geq 2D$, there exists a Lebesgue-measure zero set $N \in \mathbb{R}^d \times \ldots \mathbb{R}^d$, such that if $(x_1, \cdots, x_n) \notin N$, for any $y_i$, there is a unique $f$ (or no such $f$) to the system of equations $y_i = f(x_i)$ for $f \in \mathcal{F}_\mathcal{V}$.*

We give two important examples of admissible polynomial families with low dimension.

**Example 4.3.** (Low-rank Polynomial of rank $k$) The function $f \in \mathcal{F}_\mathcal{V}$ is a polynomial with $k$ terms, that is

$$F(x) = \sum_{i=1}^{k} \lambda_i \langle v_i, x \rangle^{p_i},$$

where $p = \max\{p_i\}$. The dimension of this family is upper bounded by $D \leq dk$. Neural network with monomial/polynomial activation functions are low-rank polynomials.

**Example 4.4.** The function $f \in \mathcal{F}_\mathcal{V}$ is of the form $f(x) = q(Ux)$, where $U \in \mathbb{R}^{k \times d}$ and $q$ is a degree $p$ polynomial. The polynomial $q$ and matrix $U$ are unknown. The dimension of this family is upper bounded by $D \leq d(k+1)^p$.

Next, we introduce the notion of positive measure.

---

[4]Admissible means the dimension of $\mathcal{X}$ decreases by one when there is an additional linear constraint $\langle \Theta, X \rangle = 0$

**Definition 4.5.** We say a measurable set $E \in \mathbb{R}^d$ is of positive measure if $\mu(E) > 0$, where $\mu$ is the standard Lebesgue measure on $\mathbb{R}^d$.

If a measurable set $E$ satisfies $\mu(E) > 0$, then there exists a procedure to draw samples from $E$, such that for any $N \subset \mathbb{R}^d$ of Lebesgue-measure zero, the probability that the sample falls in $N$ is zero. In fact, the sampling probability can be given by $\mathbb{P}_{x \in \mathcal{N}(0, I_d)}(\cdot | x \in E)$. The intuition behind its definition is that for all admissible polynomial families, the set of $(x_1, \cdots, x_n)$ with "redundant information" about learning the parameter $\Theta$ is of Lebesgue-measure zero. Therefore, a positive measure set allows you to query randomly and avoids getting coherent measurements.

Next two theorems identify the sufficent conditions for efficiently learning deterministic MDPs under polynomial realizability. Specifically, under online RL setting, we require the strong assumption that the set $\{\phi_h(s, a) | a \in \mathcal{A}\}$ is of positive measure for all $h \in [H]$ and all $s \in \mathcal{S}$, while under generative model setting, we only require the union set $\bigcup_{s \in \mathcal{S}} \{\phi_h(s, a) | a \in \mathcal{A}\}$ to be of positive measure for all $h \in [H]$. The algorithms for solving the both cases are summarized in Algorithms 4 and 5.

**Assumption 4.6** (Polynomial Realizability). For all $h \in [H]$, $Q_h^*(s_h, a_h)$, viewed as the function of $\phi_h(s_h, a_h)$, lies in some admissible polynomial family $\mathcal{F}_{\mathcal{V}_h}$ with dimension bounded by $D$.

**Theorem 4.7.** *For the generative model setting, assume that the set $\{\phi_h(s, a) \mid s \in \mathcal{S}, a \in \mathcal{A}\}$ is of positive measure at any level $h$. Under the polynomial realizability, Algorithm 4 almost surely learns the optimal policy $\pi^\star$ with at most $N = 2DH$ samples.*

**Theorem 4.8.** *For the online RL setting, assume that $\{\phi_h(s, a) \mid a \in \mathcal{A}\}$ is of positive measure for every state $s$ at every level $h$. Under polynomial realizability, within $T = 2DH$ episodes, Algorithm 5 learns the optimal policy $\pi^\star$ almost surely.*

---

**Algorithm 4** Dynamic programming under generative model settings

---

1: **for** $h = H, \cdots, 1$ **do**
2:     Sample $2D$ points $\{\phi_h(s_h^{(i)}, a_h^{(i)})\}_{i=1}^{2D}$ according to $\mathbb{P}_{x \in \mathcal{N}(0, I_d)}(\cdot \mid x \in E_h)$ where $E_h = \{\phi_h(s, a) \mid s \in \mathcal{S}, a \in \mathcal{A}\}$.
3:     Query the generative model with state-action pair $(s_h^{(i)}, a_h^{(i)})$ at level $h$ for $i = 1, \ldots, 2D$, and observe the next state $\widetilde{s}_h^{(i)}$ and reward $r_h^{(i)}$.
4:     Solve for $Q_h^*$ with the $2D$ equations $Q_h^*(s_h^{(i)}, a_h^{(i)}) = r_h^{(i)} + V_{h+1}^*(\widetilde{s}_h^{(i)})$.
5:     Set $\pi_h^*(s) = \arg\max_a Q_h^*(s, a)$ and $V_h^*(s) = \max_a Q_h^*(s, a)$.
6: **end for**
7: **Output** $\pi^*$

---

---

**Algorithm 5** Dynamic programming under online RL settings

---

1: **for** $h = H, \cdots, 1$ **do**
2:     Fix any action sequence $a_1, \cdots, a_{h-1}$.
3:     Play $a_1, \cdots, a_{h-1}$ for the first $h - 1$ levels and reach a state $s_h$. Sample $2D$ points $\{\phi_h(s_h, a_h^{(i)})\}_{i=1}^{2D}$ according to $\mathbb{P}_{x \in \mathcal{N}(0, I_d)}(\cdot \mid x \in E_h)$ where $E_h = \{\phi_h(s_h, a) \mid a \in \mathcal{A}\}$.
4:     Play $a_h^{(i)}$ at $s_h$ for $i = 1, \ldots, 2D$, and observe the next state $\widetilde{s}_h^{(i)}$ and reward $r_h^{(i)}$.
5:     Solve for $Q_h^*$ with the $2D$ equations $Q_h^*(s_h^{(i)}, a_h^{(i)}) = r_h^{(i)} + V_{h+1}^*(\widetilde{s}_h^{(i)})$.
6:     Set $\pi_h^*(s) = \arg\max_a Q_h^*(s, a)$ and $V_h^*(s) = \max_a Q_h^*(s, a)$.
7: **end for**
8: **Output** $\pi^*$

---

We remark that our Theorem 4.8 for learning MDPs under the online RL setting relies on a very strong assumption that allows the learner to explore randomly for any state. However, this assumption is necessary in some sense, as is suggested by our lower bound example in the next subsection.

## 4.1 Necessity of Generic Feature Maps in Online RL

In this section, we consider lower bounds for learning deterministic MDPs with polynomial realizable $Q^*$ under online RL setting. Our goal is to show that in the online setting the generic assumption on

the feature maps $\phi_h(s, \cdot)$ is necessary. On the contrary, under the generative model setting one can efficiently learn the MDPs without such a strong assumption, since at every level $h$ the we can set the state arbitrarily.

**MDP construction**   We briefly introduce the intuition of our construction. Consider a family of MDPs with only two states $\mathcal{S} = \{S_{\text{good}}, S_{\text{bad}}\}$. we set the feature map $\phi_h$ such that, for the good state $S_{\text{good}}$, it allows the learner to explore randomly, i.e., $\{\phi_h(S_{\text{good}}, a) \mid a \in \mathcal{A}\}$ is of postive measure.

However, for the bad state $S_{\text{bad}}$, all actions are mapped to some restricted set, which forbids random exploration, i.e., $\{\phi_h(S_{\text{bad}}, a) \mid a \in \mathcal{A}\}$ is measure zero. This is illustrated in Figure 1.

Specifically, at least $\Omega(d^p)$ actions are needed to identify the groud-truth polynomial of $Q_h^*$ for $S_{\text{bad}}$, while $O(d)$ actions suffice for $S_{\text{good}}$.

The transition $\mathbb{P}_h$ is constructed as $\mathbb{P}_h(s_{\text{bad}}|s, a) = 1$ for all $s \in \mathcal{S}, a \in \mathcal{A}$, which means it is impossible for the online scenarios to reach the good state for $h > 1$.

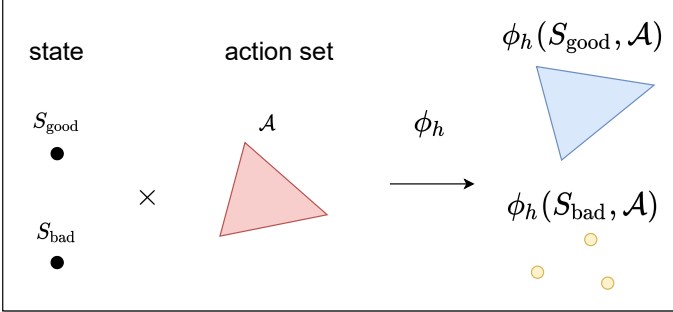

Figure 1: An illustration of the hard case for deterministic MDPs with polynomial realizable $Q^*$. The image of the feature map $\phi_h$ at $S_{\text{good}}$ is of positive measure, while the image of $\phi_h$ at $S_{\text{bad}}$ is not. This makes it difficult to learn under the online RL setting.

**Theorem 4.9.** *There exists a family of MDPs satisfying Assumption 4.6, such that the set $\{\phi_h(s, a) \mid s \in \mathcal{S}, a \in \mathcal{A}\}$ is of positive measure at any level $h$, but for all $h$ there is some $s_{\text{bad}} \in \mathcal{S}$ such that $\{\phi_h(s_{\text{bad}}, a) \mid a \in \mathcal{A}\}$ is measure zero. Under the online RL setting, any algorithm needs to play at least $\Omega(d^p)$ episodes to identify the optimal policy. On the contrary, under the generative model setting, only $O(d)H$ samples are needed.*

## 5   Conclusions

In this paper, we consider neural network and polynomial function approximations in the simulator and online settings. To our knowledge, this is the first paper that shows sample-efficient reinforcement learning is possible with neural net function approximation. Our results substantially improve upon what can be achieved with existing results that primarily rely on embedding neural networks into linear function classes. The analysis reveals that for function approximations that allows for efficient $\ell_\infty$ recovery, such as two layer neural networks and admissible polynomial families, reinforcement learning can be reduced to parameter recovery problems, as well-studied in theories for deep learning, phase retrieval, and etc. Our method can also be potentially extended to handle three-layer and deeper neural networks, with advanced tools in [23, 24].

Our results for polynomial activation require deterministic transitions, since we cannot handle how noise propagates in solving polynomial equations. We leave to future work an in-depth study of the stability of roots of polynomial systems with noise, which is a fundamental mathematical problem and even unsolved for homogeneous polynomials. In particular, noisy tensor decomposition approaches combined with zeroth-order optimization may allow for stochastic transitions [36].

In the online RL setting, we can only show efficient learning under a very strong yet necessary assumption on the feature mapping. We leave to future work identifying more realistic and natural conditions which permit efficient learning in the online RL setting.

Finally, in future work, we hope to consider deep neural networks where parameter recovery or $\ell_\infty$ error is unattainable, and deep reinforcement learning with representation learning [84, 15].

## Acknowledgements

JDL acknowledges support of the ARO under MURI Award W911NF-11-1-0303, the Sloan Research Fellowship, NSF CCF 2002272, and an ONR Young Investigator Award. QL is supported by NSF 2030859 and the Computing Research Association for the CIFellows Project. SK acknowledges funding from the NSF Award CCF-1703574 and the ONR award N00014-18-1-2247.

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
