# A Additional related work

**Deterministic RL** Deterministic system is often the starting case in the study of sample-efficient algorithms, where the issue of exploration and exploitation trade-off is more clearly revealed since both the transition kernel and reward function are deterministic. The seminal work [81] proposes a sample-efficient algorithm for Q-learning that works for a family of function classes. Recently, [21] studies the agnostic setting where the optimal Q-function can only be approximated by a function class with approximation error. The algorithm in [21] learns the optimal policy with the number of trajectories linear with the eluder dimension.

# B Omitted Proofs in Section 3

## B.1 Proof of Section 3.1

**Theorem B.1** (Formal statement of Theorem 3.4). *Consider MDP $\mathcal{M}$ where the transition is deterministic. Assume the function class in Definition 3.1 satisfies Assumption 2.1 and Assumption 2.2. For any $t \in (0,1)$, if $d \geq \Omega(\log(B_W/\lambda))$ and $n \geq d \cdot \mathrm{poly}(\kappa, k, \lambda, B_W, B_\phi, H, \log(d/t))$, then with probability at least $1-t$ Algorithm 1 returns the optimal policy $\pi^*$.*

*Proof.* Use $\pi_1^*, \ldots, \pi_H^*$ to denote the global optimal policy. We prove that Algorithm 1 learns $\pi_h^*$ from $h = H$ to $h = 1$.

At level $H$, the query obtains exact $Q_H^*(s,a)$. Therefore by Theorem B.15, $\widehat{Q}_H = Q_H^*$ and thus the optimal planning finds $\pi_H = \pi_H^*$. Suppose we have learned $\pi_{h+1}^*, \ldots, \pi_H^*$ at level $h$. Due to deterministic transition, the query obtains exact $Q_h^*(s,a)$. Therefore by Theorem B.15, $\widehat{Q}_h = Q_h^*$ and thus the optimal planning finds $\pi_h = \pi_h^*$. Recursively applying this process to $h = 1$, we complete the proof. □

## B.2 Proof of Section 3.2

**Theorem B.2** (Formal statement of Theorem 3.5). *Assume the function class in Definition 3.1 satisfies Assumption 2.1, Assumption 2.2 and is policy complete. For any $\epsilon > 0$ and $t \in (0,1)$ such that $d \geq \Omega(\log(B_W B_\phi/\epsilon))$, if $n \geq \epsilon^{-2} \cdot d \cdot \mathrm{poly}(\kappa, k, B_W, B_\phi, H, \log(d/t))$, then with probability at least $1-t$ Algorithm 2 returns a policy $\pi$ such that $V^* - V^\pi \leq \epsilon$.*

*Proof.* Use $\pi_1^*, \ldots, \pi_H^*$ to denote the global optimal policy. We prove for all $s \in \mathcal{S}$,

$$V_h^{\pi_h^*, \pi_{h+1}^*, \ldots, \pi_H^*}(s) - V_h^{\pi_h, \pi_{h+1}, \ldots, \pi_H}(s) \leq \frac{(H-h+1)\epsilon}{H}.$$

At level $H$, let $e_H(s_H^i, a_H^i) = r_H(s_H^i, a_H^i) - Q_H^*(s_H^i, a_H^i)$, then $e_H(s_H^i, a_H^i) = 0$. From Theorem B.13, we have $\widehat{Q}_H(s,a) := v_H^\top \sigma(W_H \phi(s,a))$ satisfies $|\widehat{Q}_H(s,a) - Q_H^*(s,a)| \leq \frac{\epsilon}{2H}$ for all $s \in \mathcal{S}, a \in \mathcal{A}$. Therefore for all $s \in \mathcal{S}$,

$$\begin{aligned}
V_H^*(s) - V_H^{\pi_H}(s) &= \mathbb{E}_{a \sim \pi_H^*}[Q_H^*(s,a)] - \mathbb{E}_{a \sim \pi_H^*}[\widehat{Q}_H(s,a)] \\
&\quad + \mathbb{E}_{a \sim \pi_H^*}[\widehat{Q}_H(s,a)] - \mathbb{E}_{a \sim \pi_H}[\widehat{Q}_H(s,a)] \\
&\quad + \mathbb{E}_{a \sim \pi_H}[\widehat{Q}_H(s,a)] - \mathbb{E}_{a \sim \pi_H}[Q_H^*(s,a)] \\
&\leq \frac{\epsilon}{H}
\end{aligned}$$

where in the second step we used $\mathbb{E}_{a \sim \pi_H^*}[\widehat{Q}_H(s,a)] \leq \mathbb{E}_{a \sim \pi_H}[\widehat{Q}_H(s,a)]$ by optimality of $\pi_H$ and $|\widehat{Q}_H(s,a) - Q_H^*(s,a)| \leq \frac{\epsilon}{2H}$.

Suppose we have learned policies $\pi_{h+1}, \ldots, \pi_H$, we use $\widetilde{\pi}_h$ to denote the optimal policy of $Q_h^{\pi_{h+1}, \ldots, \pi_H}(s,a)$. Let

$$e_h(s_h^i, a_h^i) = \widehat{Q}_h^i - Q_h^{\pi_{h+1}, \ldots, \pi_H}(s_h^i, a_h^i)$$

then $e_h(s_h^i, a_h^i)$ is zero mean $H^2$ sub-Gaussian (notice that $\widehat{Q}_h^i$ is unbiased estimate of $Q_h^{\pi_{h+1},\ldots,\pi_H}(s_h^i, a_h^i)$, and $\widehat{Q}_h^i \leq O(H)$). From Theorem B.13, we have $\widehat{Q}_h(s,a) = v_h^\top \sigma(W_h \phi(s,a))$ satisfies $|\widehat{Q}_h(s,a) - Q_h^{\pi_{h+1},\ldots,\pi_H}(s,a)| \leq \frac{\epsilon}{2H}$ for all $s \in \mathcal{S}, a \in \mathcal{A}$. Therefore for all $s \in \mathcal{S}$,

$$V_h^{\widetilde{\pi}_h, \pi_{h+1}, \ldots, \pi_H}(s) - V_h^{\pi_h, \pi_{h+1}, \ldots, \pi_H}(s)$$
$$= \mathbb{E}_{a \sim \widetilde{\pi}_h}[Q_h^{\pi_{h+1},\ldots,\pi_H}(s,a)] - \mathbb{E}_{a \sim \widetilde{\pi}_h}[\widehat{Q}_h(s,a)]$$
$$+ \mathbb{E}_{a \sim \widetilde{\pi}_h}[\widehat{Q}_h(s,a)] - \mathbb{E}_{a \sim \pi_h}[\widehat{Q}_h(s,a)]$$
$$+ \mathbb{E}_{a \sim \pi_h}[\widehat{Q}_h(s,a)] - \mathbb{E}_{a \sim \pi_h}[Q_h^{\pi_h, \pi_{h+1}, \ldots, \pi_H}(s,a)]$$
$$\leq \frac{\epsilon}{H}$$

where in the second step we used $\mathbb{E}_{a \sim \widetilde{\pi}_h}[\widehat{Q}_h(s,a)] \leq \mathbb{E}_{a \sim \pi_h}[\widehat{Q}_h(s,a)]$ by optimality of $\pi_h$ and $|\widehat{Q}_h(s,a) - Q_h^{\pi_{h+1},\ldots,\pi_H}(s,a)| \leq \frac{\epsilon}{2H}$.

It thus follows that

$$V_h^{\pi_h^*, \pi_{h+1}^*, \ldots, \pi_H^*}(s) - V_h^{\pi_h, \pi_{h+1}, \ldots, \pi_H}(s) = V_h^{\pi_h^*, \pi_{h+1}^*, \ldots, \pi_H^*}(s) - V_h^{\pi_h^*, \pi_{h+1}, \ldots, \pi_H}(s)$$
$$+ V_h^{\pi_h^*, \pi_{h+1}, \ldots, \pi_H}(s) - V_h^{\widetilde{\pi}_h, \pi_{h+1}, \ldots, \pi_H}(s)$$
$$+ V_h^{\widetilde{\pi}_h, \pi_{h+1}, \ldots, \pi_H}(s) - V_h^{\pi_h, \pi_{h+1}, \ldots, \pi_H}(s)$$
$$\leq V_h^{\pi_h^*, \pi_{h+1}^*, \ldots, \pi_H^*}(s) - V_h^{\pi_h^*, \pi_{h+1}, \ldots, \pi_H}(s) + \frac{\epsilon}{H}$$
$$\leq \cdots$$
$$\leq \frac{(H-h+1)\epsilon}{H}.$$

where in the second step we use $V_h^{\pi_h^*, \pi_{h+1}, \ldots, \pi_H}(s) \leq V_h^{\widetilde{\pi}_h, \pi_{h+1}, \ldots, \pi_H}(s)$ from optimality of $\widetilde{\pi}_h$. Repeating this argument to $h = 1$ completes the proof $\qquad \square$

### B.3 Proof of Section 3.3

**Theorem B.3** (Formal statement of Theorem 3.6). *Assume the function class in Definition 3.1 satisfies Assumption 2.1, Assumption 2.2, and is Bellman complete. For any $\epsilon > 0$ and $t \in (0,1)$ such that $d \geq \Omega(\log(B_W B_\phi/\epsilon))$, if $n \geq \epsilon^{-2} \cdot d \cdot \mathrm{poly}(\kappa, k, B_W, B_\phi, H, \log(d/t))$, then with probability at least $1 - t$ Algorithm 3 returns a policy $\pi$ such that $V^* - V^\pi \leq \epsilon$.*

*Proof.* Use $\pi_1^*, \ldots, \pi_H^*$ to denote the global optimal policy. We prove

$$|\widehat{Q}_h(s,a) - Q_h^*(s,a)| \leq \frac{(H-h+1)\epsilon}{H} \tag{1}$$

for all $s \in \mathcal{S}, a \in \mathcal{A}$.

At level $H$, let

$$e_H(s_H^i, a_H^i) = r_H(s_H^i, a_H^i) - Q_H^*(s_H^i, a_H^i)$$

then $e_H(s_H^i, a_H^i) = 0$. From Theorem B.13, we have $\widehat{Q}_H(s,a) := v_H^\top \sigma(W_H \phi(s,a))$ satisfies $|\widehat{Q}_H(s,a) - Q_H^*(s,a)| \leq \frac{\epsilon}{H}$ for all $s \in \mathcal{S}, a \in \mathcal{A}$.

Suppose we have learned $\widehat{Q}_{h+1}(s,a)$ with $|\widehat{Q}_{h+1}(s,a) - Q_{h+1}^*(s,a)| \leq \frac{(H-h)\epsilon}{H}$. At level $h$, let

$$e_h(s_h^i, a_h^i) = r_h(s_h^i, a_h^i) + \widehat{V}_{h+1}(s_{h+1}^i) - \mathcal{T}_h(\widehat{Q}_{h+1})(s_h^i, a_h^i)$$

then $e_h(s_h^i, a_h^i)$ is zero mean $H^2$ sub-Gaussian (notice that $r_h(s_h^i, a_h^i) + \widehat{V}_{h+1}(s_{h+1}^i)$ is unbiased estimate of $\mathcal{T}_h(\widehat{Q}_{h+1})(s_h^i, a_h^i)$, and $r_h(s_h^i, a_h^i) + \widehat{V}_{h+1}(s_{h+1}^i) \leq O(H)$). From Theorem B.13, we have $\widehat{Q}_h(s,a) := v_h^\top \sigma(W_h \phi(s,a))$ satisfies $|\widehat{Q}_h(s,a) - \mathcal{T}_h(\widehat{Q}_{h+1})(s_h^i, a_h^i)| \leq \frac{\epsilon}{H}$ for all $s \in \mathcal{S}, a \in$

$\mathcal{A}$. Therefore

$$|\widehat{Q}_h(s,a) - Q_h^*(s,a)| \le |\widehat{Q}_h(s,a) - \mathcal{T}_h(\widehat{Q}_{h+1})(s,a)| + |\mathcal{T}_h(\widehat{Q}_{h+1})(s,a) - Q_h^*(s,a)|$$

$$\le \frac{\epsilon}{H} + \max_{s \in \mathcal{S}, a \in \mathcal{A}} |\widehat{Q}_{h+1}(s,a) - Q_{h+1}^*(s,a)|$$

$$\le \frac{(H - h + 1)\epsilon}{H}$$

holds for all $s \in \mathcal{S}, a \in \mathcal{A}$.

It thus follows that for all $s_1 \in \mathcal{S}$,

$$V_h^{\pi_1^*,\ldots,\pi_H^*}(s_1) - V_h^{\pi_1,\ldots,\pi_H}(s_1) = \mathbb{E}_{a \sim \pi_1^*}[Q_1^*(s_1,a)] - \mathbb{E}_{a \sim \pi_1}[Q_1^{\pi_2,\ldots,\pi_H}(s_1,a)]$$

$$\le \mathbb{E}_{a \sim \pi_1^*}[\widehat{Q}_1(s_1,a)] - \mathbb{E}_{a \sim \pi_1}[Q_1^{\pi_2,\ldots,\pi_H}(s_1,a)] + \epsilon$$

$$\le \mathbb{E}_{a \sim \pi_1}[\widehat{Q}_1(s_1,a) - Q_1^{\pi_2,\ldots,\pi_H}(s_1,a)] + \epsilon$$

$$\le \mathbb{E}_{a \sim \pi_1}[Q_1^*(s_1,a) - Q_1^{\pi_2,\ldots,\pi_H}(s_1,a)] + 2\epsilon$$

$$\le \mathbb{E}_{a \sim \pi_1}\mathbb{E}_{s_2 \sim \mathbb{P}(\cdot|s,a)}[V_2^{\pi_2^*,\ldots,\pi_H^*}(s_2) - V_2^{\pi_2,\ldots,\pi_H}(s_2)] + 2\epsilon$$

$$\le \cdots$$

$$\le 2H\epsilon$$

where the first step comes from definition of value function; the second step comes from Eq (1); the third step comes from optimality of $\pi_1$; the fourth step comes from Eq (1); the fifth step comes from Bellman equation. The proof is complete by rescaling $\epsilon \leftarrow \epsilon/H$. □

### B.4 Proof of Section 3.4

With gap condition, either Algorithm 2 or Algorithm 3 will work as long as we select $\epsilon \approx \rho$. The following displays an adaption from Algorithm 2.

---

**Algorithm 6** Learning realizable $Q^*$ with optimality gap

---

1: **for** $h = H, \ldots 1$ **do**
2:     Sample $x_h^i, i \in [n]$ from standard Gaussian $\mathcal{N}(0, I_d)$
3:     **for** $i \in [n]$ **do**
4:         **if** $\|x_h^i\| \le \delta_\phi$ **then**
5:             Find $(s_h^i, a_h^i) \in \phi^{-1}(x_h^i)$ and locate the state $s_h^i$ in the generative model
6:             Pull action $a_h^i$ and use $\pi_{h+1}, \ldots, \pi_H$ as the roll-out to collect rewards $r_h^{(i)}, \ldots, r_H^{(i)}$
7:             Construct unbiased estimation of $Q_h^{\pi_{h+1},\ldots,\pi_H}(s_h^i, a_h^i)$

$$\widehat{Q}_h^i \leftarrow r_h^{(i)} + \cdots + r_H^{(i)}$$

8:         **else**
9:             Let $\widehat{Q}_h^i \leftarrow 0$.
10:         **end if**
11:     **end for**
12:     Compute $(v_h, W_h) \leftarrow \text{NEURALNETNOISYRECOVERY}(\{(x_h^i, \widehat{Q}_h^i) : i \in [n]\})$
13:     Set $\widehat{Q}_h(s,a) \leftarrow v_h^\top \sigma(W_h \phi(s,a))$
14:     Let $\pi_h(s) \leftarrow \arg\max_{a \in \mathcal{S}} \widehat{Q}_h(s,a)$
15: **end for**
16: **Return** $\pi_1, \ldots, \pi_H$

---

**Theorem B.4** (Formal statement of Theorem 3.8). *Assume the function class in Definition 3.1 satisfies Assumption 2.1 and Assumption 2.2. Suppose $\rho > 0$ and $d \ge \Omega(\log(B_W B_\phi/\rho))$, for any $t \in (0,1)$, if $n = \frac{d}{\rho^2} \cdot \text{poly}(\kappa, k, B_W, B_\phi, H, \log(d/t))$, then with probability at least $1 - t$ Algorithm 6 returns the optimal policy $\pi^*$.*

*Proof.* Use $\pi_1^*, \ldots, \pi_H^*$ to denote the global optimal policy. Similar to Theorem B.1, we prove that Algorithm 6 learns $\pi_h^*$ from $h = H$ to $h = 1$.

At level $H$, the algorithm uses $n = \frac{d}{\rho^2} \cdot \text{poly}(\kappa, k, \log d, B_W, B_\phi, H, \log(1/t))$ trajectories to obtain $\widehat{Q}_H$ such that $|\widehat{Q}_H(s, a) - Q^*(s, a)| \leq \rho/4$ by Theorem B.13. Therefore

$$
\begin{aligned}
&V_H^*(s) - Q_H^*(s, \pi_H(s)) \\
&\leq Q_H^*(s, \pi_H^*(s)) - Q_H^*(s, \pi_H(s)) \\
&\leq Q_H^*(s, \pi_H^*(s)) - \widehat{Q}_H(s, \pi_H^*(s)) + \widehat{Q}_H(s, \pi_H^*(s)) - \widehat{Q}_H(s, \pi_H(s)) \\
&\quad + \widehat{Q}_H(s, \pi_H(s)) - Q_H^*(s, \pi_H(s)) \\
&\leq \rho/2
\end{aligned}
$$

where the third inequality uses the optimality of $\pi_H(s)$ under $\widehat{Q}_H$. Thus Definition 3.7 gives $\pi_H(s) = \pi_H^*(s)$. Suppose we have learned $\pi_{h+1}^*, \ldots, \pi_H^*$ at level $h$. We apply the same argument to derive $\pi_h = \pi_h^*$. Recursively applying this process to $h = 1$, we complete the proof. $\qquad\square$

## B.5 Neural network recovery

This section considers recovering neural network $\langle v, \sigma(Wx) \rangle$ from the following two models, where $B = \Omega(d \cdot \text{poly} \log(d))$.

- Noisy samples from

$$
x \sim \mathcal{N}(0, I_d), \quad y = (\langle v, \sigma(Wx) \rangle + \xi) \cdot \mathbb{1}(\|x\| \leq B) \tag{2}
$$

  where $\xi$ is $\vartheta$ sub-Gaussian noise.
- Noiseless samples from

$$
x \sim \mathcal{N}(0, I_d), \quad y = (\langle v, \sigma(Wx) \rangle) \cdot \mathbb{1}(\|x\| \leq B) \tag{3}
$$

Recovering neural network has received comprehensive study in deep learning theory [39, 90, 27]. The analysis in this section is mainly based on the method of moments in [90]. However, notice that the above learning tasks are different from those considered in [90], due to the presence of noise and the truncated signals. Therefore, additional considerations must be made in the analysis.

We consider more general homogeneous activation functions, specified by the assumptions that follow. Since the activation function is homogeneous, we assume $v_i \in \{\pm 1\}$ in the following without loss of generality.

**Assumption B.5** (Property 3.1 of [90]). Assume $\sigma'(x)$ is nonnegative and homogeneously bounded, i.e. $0 \leq \sigma'(x) \leq L_1 |x|^p$ for some constants $L_1 > 0$ and $p \geq 0$.

**Definition B.6** (Part of property 3.2 of [90]). Define $\rho(z) := \min\{\beta_0(z) - \alpha_0^2(z) - \alpha_1^2(z), \beta_2(z) - \alpha_1^2(z) - \alpha_2^2(z), \alpha_0(z)\alpha_2(z) - \alpha_1^2(z)\}$, where $\alpha_q(z) := \mathbb{E}_{x \sim \mathcal{N}(0,1)}[\sigma'(zx)x^q], q \in \{0, 1, 2\}$, and $\beta_q(z) := \mathbb{E}_{x \sim \mathcal{N}(0,1)}[(\sigma')^2(zx)x^q]$ for $q \in \{0, 2\}$.

**Assumption B.7** (Part of property 3.2 of [90]). The first derivative $\sigma'(z)$ satisfies that, for all $z > 0$, we have $\rho(z) > 0$.

**Assumption B.8** (Property 3.3 of [90]). The second derivative $\sigma''(x)$ is either **(a)** globally bounded or **(b)** $\sigma''(x) = 0$ except for finite points.

Notice that ReLU, squared ReLU, leaky ReLU, and polynomial activation function functions all satisfies the above assumption. We make the following assumption on the dimension of feature vectors, which corresponds to how features can extract information about neural networks from noisy samples. The dimension only has to be greater than a logarithmic term in $1/\epsilon$ and the norm of parameters.

**Assumption B.9** (Rich feature). Assume $d \geq \Omega(\log(B_W/\epsilon))$.

First we introduce a notation from [90].

**Definition B.10.** Define outer product $\widetilde{\otimes}$ as follows. For a vector $v \in \mathbb{R}^d$ and an identity matrix $I \in \mathbb{R}^{d \times d}$,

$$
v \widetilde{\otimes} I = \sum_{j=1}^d [v \otimes e_j \otimes e_j + e_j \otimes v \otimes e_j + e_j \otimes e_j \otimes v].
$$

For a symmetric rank-$r$ matrix $M = \sum_{i=1}^{r} s_i v_i v_i^{\top}$ and an identity matrix $I \in \mathbb{R}^{d \times d}$,

$$M \widetilde{\otimes} I = \sum_{i=1}^{r} s_i \sum_{j=1}^{d} \sum_{l=1}^{6} A_{l,i,j}$$

where $A_{1,i,j} = v_i \otimes v_i \otimes e_j \otimes e_j$, $A_{2,i,j} = v_i \otimes e_j \otimes v_i \otimes e_j$, $A_{3,i,j} = e_j \otimes v_i \otimes v_i \otimes e_j$, $A_{4,i,j} = v_i \otimes e_j \otimes e_j \otimes v_i$, $A_{5,i,j} = e_j \otimes v_i \otimes e_j \otimes v_i$, $A_{6,i,j} = e_j \otimes e_j \otimes v_i \otimes v_i$.

Now we define some moments.

**Definition B.11.** Define $M_1, M_2, M_3, M_4, m_{1,i}, m_{2,i}, m_{3,i}, m_{4,i}$ as follows:

$$
\begin{aligned}
M_1 &:= \mathbb{E}[y \cdot x] \\
M_2 &:= \mathbb{E}[y \cdot (x \otimes x - I)] \\
M_3 &:= \mathbb{E}[y \cdot (x^{\otimes 3} - x \widetilde{\otimes} I)] \\
M_4 &:= \mathbb{E}[y \cdot (x^{\otimes 4} - (x \otimes x) \widetilde{\otimes} I + I \widetilde{\otimes} I)] \\
\gamma_j(x) &:= \mathbb{E}_{z \sim \mathcal{N}(0,1)}[\sigma(x \cdot z) z^j], \forall j \in 0, 1, 2, 3, 4 \\
m_{1,i} &:= \gamma_1(\|w_i\|) \\
m_{2,i} &:= \gamma_2(\|w_i\|) - \gamma_0(\|w_i\|) \\
m_{3,i} &:= \gamma_3(\|w_i\|) - 3\gamma_1(\|w_i\|) \\
m_{4,i} &:= \gamma_4(\|w_i\|) + 3\gamma_0(\|w_i\|) - 6\gamma_2(\|w_i\|)
\end{aligned}
$$

The above expectations are all with respect to $x \sim \mathcal{N}(0, I_d)$ and $y = \langle v, \sigma(Wx) \rangle$.

**Assumption B.12** (Assumption 5.3 of [90]). Assume the activation function satisfies the followings:

- If $M_i \neq 0$, then $m_{j,i} \neq 0$ for all $i \in [k]$.

- At least one of $M_3$ and $M_4$ is not zero.

- If $M_1 = M_3 = 0$, then $\sigma(z)$ is an even function.

- If $M_2 = M_4 = 0$, then $\sigma(z)$ is an odd function.

Now we state the theoretical result that recovers neural networks from noisy data.

**Theorem B.13** (Neural network recovery from noisy data). *Let the activation function $\sigma$ satisfies Assumption B.5 and Assumption B.12. Let $\kappa$ be the condition number of $W$. Given $n$ samples from Eq (2). For any $t \in (0, 1)$ and $\epsilon \in (0, 1)$ such that Assumption B.9 holds, if*

$$n \geq \epsilon^{-2} \cdot d \cdot \mathrm{poly}(\kappa, k, \vartheta, \log(d/t))$$

*then there exists an algorithm that takes $\widetilde{O}(nkd)$ time and returns a matrix $\widehat{W} \in \mathbb{R}^{k \times d}$ and a vector $\widehat{v} \in \{\pm 1\}^k$ such that with probability at least $1 - t$,*

$$\|\widehat{W} - W\|_F \leq \epsilon \cdot \mathrm{poly}(k, \kappa) \cdot \|W\|_F, \text{ and } \widehat{v} = v.$$

The algorithm and proof are shown in Appendix B.5.1. By Assumption B.5, the following corollary is therefore straightforward.

**Corollary B.14.** *In the same setting as Theorem B.13. For any $t \in (0, 1)$ and suppose $\|W\|_F \leq B_W$ and Assumption B.9 holds. Given $n$ samples from Eq (2). If*

$$n \geq \epsilon^{-2} \cdot d \cdot \mathrm{poly}(\kappa, k, \log(d/t), B_W, B_\phi, \vartheta)$$

*then there exists an algorithm that takes $\widetilde{O}(nkd)$ time and outputs a matrix $\widehat{W} \in \mathbb{R}^{k \times d}$ and a vector $\widehat{v} \in \{\pm 1\}^k$ such that with probability at least $1 - t$, for all $\|x\|_2 \leq B_\phi$*

$$|\langle \widehat{v}, \sigma(\widehat{W}x) \rangle - \langle v, \sigma(Wx) \rangle| \leq \epsilon.$$

*In particular, when $B_\phi = O(d \cdot \mathrm{poly} \log d)$ the following sample complexity suffices*

$$n \geq \epsilon^{-2} \cdot d^{O(1+p)} \cdot \mathrm{poly}(\kappa, k, \log(d/t), B_W, \vartheta).$$

Now we state the theoretical result that precisely recovers neural networks from noiseless data. The proof and method are shown in Appendix B.5.2.

**Theorem B.15** (Exact neural network recovery from noiseless data)**.** *Let the activation function satisfies Assumption B.5 and Assumption B.12, Assumption B.7 and Assumption B.8(b). Given $n$ samples from Eq (3). For any $t \in (0,1)$, suppose $d \geq \Omega(\log(B_W/\lambda))$ and*

$$n \geq d \cdot \mathrm{poly}(\kappa, k, \lambda, \log(d/t)),$$

*then there exists an algorithm that output exact $W$ and $v$ with probability at least $1 - t$.*

### B.5.1 Recover neural networks from noisy data

In this section we prove Theorem B.13. Denote $W = [w_1, \cdots, w_k]^\top$ where $w_i \in \mathbb{R}^d$ and $\overline{w_i} = w_i/\|w_i\|_2$.

**Definition B.16.** Given a vector $\alpha \in \mathbb{R}^d$. Define $P_2 := M_{j_2}(I, I, \alpha, \cdots, \alpha)$ where $j_2 = \min\{j \geq 2 : M_j \neq 0\}$ and $P_3 := M_{j_3}(I, I, I, \alpha, \cdots, \alpha)$ where $j_3 = \min\{j \geq 3 : M_j \neq 0\}$.

The method of moments is presented in Algorithm 7. Here we sketch its ideas, and refer readers to [90] for thorough explanations. There are three main steps. In the first step, it computes the span of the rows of $W$. By power method, Line 7 finds the top-$k$ eigenvalues of $CI + \widehat{P}_2$ and $CI - \widehat{P}_2$. It then picks the largest $k$ eigenvalues from $CI + \widehat{P}_2$ and $CI - \widehat{P}_2$, by invoking TOPK in Line 15. Finally it orthogonalizes the corresponding eigenvectors in Line 19 and finds an orthogonal matrix $V$ in the subspace spanned by $\{\overline{w}_1, \ldots, \overline{w}_k\}$.

In the second step, the algorithm forms third order tensor $R_3 = P_s(V, V, V) \in \mathbb{R}^{k \times k \times k}$ and use the robust tensor decomposition method in [47] to find $\widehat{u}$ that approximates $s_i V^\top \overline{w}_i$ with unknown signs $s_i$. In the third step, the algorithm determines $s$, $v$ and $w_i$, $i \in [k]$. Since the activation function is homogeneous, we assume $v_i \in \{\pm 1\}$ and $m_{j,i} = c_j\|w_i\|^{p+1}$ for universal constants $c_j$ without loss of generality. For illustration, we define $Q_1$ and $Q_2$ as follows.

$$Q_1 = M_{l_1}(I, \underbrace{\alpha, \cdots, \alpha}_{(l_1-1)\ \alpha\text{'s}}) = \sum_{i=1}^{k} v_i c_{l_1} \|w_i\|^{p+1}(\alpha^\top \overline{w}_i)^{l_1-1}\overline{w}_i, \tag{4}$$

$$Q_2 = M_{l_2}(V, V, \underbrace{\alpha, \cdots, \alpha}_{(l_2-2)\ \alpha\text{'s}}) = \sum_{i=1}^{k} v_i c_{l_2} \|w_i\|^{p+1}(\alpha^\top \overline{w}_i)^{l_2-2}(V^\top \overline{w}_i)(V^\top \overline{w}_i)^\top, \tag{5}$$

where $l_1 \geq 1$ such that $M_{l_1} \neq 0$ and $l_2 \geq 2$ such that $M_{l_2} \neq 0$ are specified later. Then the solutions of the following linear systems

$$z^* = \arg\min_{z \in \mathbb{R}^k} \left\| \sum_{i=1}^{k} z_i s_i \overline{w}_i - Q_1 \right\|, \quad r^* = \arg\min_{r \in \mathbb{R}^k} \left\| \sum_{i=1}^{k} r_i V^\top \overline{w}_i (V^\top \overline{w}_i)^\top - Q_2 \right\|_F. \tag{6}$$

are the followings

$$z_i^* = v_i s_i^{l_1} c_{l_1} \|w_i\|^{p+1}(\alpha^\top s_i \overline{w}_i)^{l_1-1}, \quad r_i = v_i s_i^{l_2} c_{l_2} \|w_i\|^{p+1}(\alpha^\top s_i \overline{w}_i)^{l_2-2}.$$

When $c_{l_1}$ and $c_{l_2}$ do not have the same sign, we can recover $v_i$ and $s_i$ by $v_i = \mathrm{sign}(r_i^* c_{l_2})$, $s_i = \mathrm{sign}(v_i z_i^* c_{l_1})$, and recover $w_i$ by

$$w_i = \left( \left| \frac{z_i^*}{c_{l_1}(\alpha^\top s_i \overline{w}_i)^{l_1-1}} \right| \right)^{1/(p+1)} \overline{w}_i.$$

In Algorithm 7, we use $V\widehat{u}_i$ to approximate $s_i\overline{w}_i$, and use moment estimators $\widehat{Q}_1$ and $\widehat{Q}_2$ to approximate $Q_1$ and $Q_1$. Then the solutions $\widehat{z}, \widehat{r}$ to the optimization problems in Line 29 should approximate $z^*$ and $r^*$, due to robustness for solving linear systems. As such, the outputs $\widetilde{v}, \widetilde{W}$ approximately recover the true model parameters.

Since Algorithm 7 carries out the same computation as [90], the computational complexity is the same. The difference of sample complexity comes from the noise $\xi$ in the model and the truncation of standard Gaussian. The proof entails bounding the error in estimating $P_2$ in Line 4, $R_3$ in Line 20 and $Q_1, Q_2$ in Line 28. In the following, unless further specified, the expectations are all with respect to $x \sim \mathcal{N}(0, I_d)$ and $y \sim (\langle v, \sigma(Wx) \rangle + \xi) \cdot \mathbb{1}(\|x\| \leq B)$.

---

**Algorithm 7** Using method of moments to recover neural network parameters

---

1: **procedure** NEURALNETNOISYRECOVERY($S = \{(x_i, y_i) : i \in [n]\}$)
2:      Choose $\alpha$ to be a random unit vector
3:      Partition $S$ into $S_1, S_2, S_3, S_4$ of equal size
4:      $\widehat{P}_2 \leftarrow \mathbb{E}_{S_1}[P_2], C \leftarrow 3\|P_2\|, T \leftarrow \log(1/\epsilon)$
5:      Choose $\widehat{V}_1^{(0)}, \widehat{V}_1^{(0)} \in \mathbb{R}^{d \times k}$ to be random matrices             ▷ Estimate subspace $V$
6:      **for** $t = 1, \ldots, T$ **do**
7:          $\widehat{V}_1^{(t)} \leftarrow \mathrm{QR}(C\widehat{V}_1^{(t-1)} + \widehat{P}_2\widehat{V}_1^{(t-1)}), \widehat{V}_2^{(t)} \leftarrow \mathrm{QR}(C\widehat{V}_2^{(t-1)} - \widehat{P}_2\widehat{V}_2^{(t-1)})$
8:      **end for**
9:      **for** j = 1,2 **do**
10:          $\widehat{V}_1^{(T)} \leftarrow [\widehat{V}_{j,1}, \cdots, \widehat{V}_{j,k}]$
11:          **for** $i \in [k]$ **do**
12:              $\lambda_{j,i} \leftarrow |\widehat{V}_{j,i}\widehat{P}_2\widehat{V}_{j,i}|$
13:          **end for**
14:      **end for**
15:      $\pi_1, \pi_2, k_1, k_2 \leftarrow \mathrm{TOPK}(\lambda, k)$
16:      **for** j = 1,2 **do**
17:          $V_j \leftarrow [\widehat{V}_{j,\pi_j(1)}, \cdots, \widehat{V}_{j,\pi_j(k_j)}]$
18:      **end for**
19:      $\widetilde{V}_2 \leftarrow \mathrm{QR}((I - V_1 V_1^\top)V_2), V \leftarrow [V_1, \widetilde{V}_2]$
20:      $\widehat{R}_3 \leftarrow \mathbb{E}_{S_2}[P_3(V, V, V)], \{\widehat{u}_i\}_{i \in [k]} \leftarrow \mathrm{TENSORDECOMPOSITION}(\widehat{R}_3)$     ▷ Learn $s_i V^\top \overline{w}_i$
21:      **if** $M_1 = M_3 = 0$ **then**
22:          $l_1, l_2 = \min\{j \in \{2, 4\} : M_j \neq 0\}$
23:      **else if** $M_2 = M_4 = 0$ **then**
24:          $l_1 \leftarrow \min\{j \in \{1, 3\} : M_j \neq 0\}, l_2 \leftarrow 3$
25:      **else**
26:          $l_1 \leftarrow \min\{j \in \{1, 3\} : M_j \neq 0\}, l_2 = \min\{j \in \{2, 4\} : M_j \neq 0\}$
27:      **end if**
28:      $\widehat{Q}_1 \leftarrow \mathbb{E}_{S_3}[Q_1], \widehat{Q}_2 \leftarrow \mathbb{E}_{S_4}[Q_2]$
29:      $\widehat{z} \leftarrow \arg\min_z \|\sum_{i=1}^k z_i V\widehat{u}_i - \widehat{Q}_1\|, \widehat{r} \leftarrow \arg\min_r \|\sum_{i=1}^k r_i\widehat{u}_i\widehat{u}_i - \widehat{Q}_2\|_F$
30:      **for** $i = 1, \ldots, k$ **do**                    ▷ Learn parameters $v, W$
31:          $\widehat{v}_i \leftarrow \mathrm{sign}(\widehat{r}_i c_{l_2}), \widehat{s}_i \leftarrow \mathrm{sign}(\widehat{v}_i\widehat{z}_i c_{l_1})$
32:          $\widehat{w}_i \leftarrow \widehat{s}_i(|\frac{\widehat{z}_i}{c_{l_1}(\alpha^\top V\widehat{u}_i)^{l_1-1}}|)^{1/(p+1)}V\widehat{u}_i$
33:      **end for**
34:      $\widehat{W} \leftarrow [\widehat{w}_i, \cdots, \widehat{w}_k]$
35:      **Return** $(\widehat{v}, \widehat{W})$
36: **end procedure**

---

**Lemma B.17.** *Let $\widehat{P}_2$ be computed in Line 4 of Algorithm 7 and $P_2$ defined in Definition B.16. Suppose $m_0 = \min_{i\in[k]}\{|m_{j_2,i}|^2(\overline{w}_i^\top\alpha)^{2(j_2-2)}\}$ and*

$$|S| \gtrsim d \cdot \mathrm{poly}(\kappa, \vartheta, \log(d/t))/(\epsilon^2 m_0)$$

*then with probability at least $1 - t$,*

$$\|P_2 - \widehat{P}_2\| \lesssim \epsilon \sum_{i=1}^k |v_i m_{j_2,i}(\overline{w}_i^\top\alpha)^{j_2-2}| + \epsilon.$$

*Proof.* It suffices to bound $\|M_2 - \widehat{M}_2\|$, $\|M_3(I, I, \alpha) - \widehat{M}_3(I, I, \alpha)\|$ and $\|M_4(I, I, \alpha, \alpha) - \widehat{M}_4(I, I, \alpha, \alpha)\|$. The main strategy is to bound all relevant moment terms and to invoke Claim E.6. Specifically, we show that with probability at least $1 - t/4$,

$$\|M_2 - \widehat{M}_2\| \lesssim \epsilon \sum_{i=1}^k |v_i m_{2,i}| + \epsilon. \tag{7}$$

$$\|M_3(I, I, \alpha) - \widehat{M}_3(I, I, \alpha)\| \lesssim \epsilon \sum_{i=1}^{k} |v_i m_{3,i}(\overline{w}_i^\top \alpha)| + \epsilon. \tag{8}$$

$$\|M_4(I, I, \alpha, \alpha) - \widehat{M}_4(I, I, \alpha, \alpha)\| \lesssim \epsilon \sum_{i=1}^{k} |v_i m_{4,i}|(\overline{w}_i^\top \alpha)^2 + \epsilon. \tag{9}$$

Recall that for sample $(x_j, y_j) \in S$, $y_j = \sum_{i=1}^{k} v_i \sigma(w_i^\top x_j) + \xi_j$ where $\xi_j$ is independent of $x_j$. Consider each component $i \in [k]$. Define $C_i(x_j), B_i(x_j) \in \mathbb{R}^{d \times d}$ as follows:

$$B_i(x_j) = (\sigma(w_i^\top x_j) + \xi_j) \cdot (x_j^{\otimes 4} - (x_j \otimes x_j) \widetilde{\otimes} I + I \widetilde{\otimes} I)(I, I, \alpha, \alpha)$$
$$= (\sigma(w_i^\top x_j) + \xi_j) \cdot ((x^\top \alpha)^2 x^{\otimes 2} - (\alpha^\top x)^2 I - 2(\alpha^\top x)(x\alpha^\top + \alpha x^\top) - xx^\top + 2\alpha\alpha^\top + I),$$

and $C_i(x_j) = \mathbb{1}(\|x_j\| \le B) \cdot B_i(x_j)$. Then from Claim E.5 we have $\mathbb{E}[B_i(x_j)] = m_{4,i}(\overline{w}_i^\top \alpha)^2 \overline{w}_i \overline{w}_i^\top$. We calculate

$$\sigma(w_i^\top x_j) \cdot (x_j^{\otimes 4} - (x_j \otimes x_j) \widetilde{\otimes} I + I \widetilde{\otimes} I)(I, I, \alpha, \alpha)$$
$$\lesssim (|w_i^\top x_j|^{p+1} + |\phi(0)|) \cdot ((x_j^\top \alpha)^2 \|x_j\|^2 + 1 + \|x_j\|^2 + (\alpha^\top x_j)^2)$$
$$\lesssim |w_i|^{p+1} \cdot |x_j|^{p+5},$$

By Assumption B.5, using Claim E.1 and $B \ge d \cdot \text{poly} \log(d)$ we have

$$\|\mathbb{E}[C_i(x_j)] - m_{4,i}(\overline{w}_i^\top \alpha)^2 \overline{w}_i \overline{w}_i^\top \| \lesssim \mathbb{E}[\mathbb{1}_{\|x_j\| \ge B} |w_i|^{p+1} \cdot |x_j|^{p+5}]$$
$$\lesssim (\|w_i\| d)^{p+5} \cdot e^{-\Omega(d \log d)}$$
$$\lesssim \epsilon.$$

Also, $\frac{1}{2}|m_{4,i}|(\overline{w}_i^\top \alpha)^2 \le \|\mathbb{E}[C_i(x_j)]\| \le 2|m_{4,i}|(\overline{w}_i^\top \alpha)^2$.

For any constant $t \in (0, 1)$, we have with probability $1 - t/4$,

$$\|C_i(x_j)\| \lesssim (|w_i^\top x_j|^{p+1} + |\phi(0)| + |\xi_j|) \cdot ((x_j^\top \alpha)^2 \|x_j\|^2 + 1 + \|x_j\|^2 + (\alpha^\top x_j)^2)$$
$$\lesssim (\|w_i\|^{p+1} + |\phi(0)| + \vartheta) \cdot d \cdot \text{poly}(\log(d/t))$$

where the first step comes from Assumption B.5 and the second step comes from Claim E.2 and Claim E.3.

Using Claim E.4, we have

$$\|\mathbb{E}[C_i(x_j)^2]\| \lesssim (\mathbb{E}[(\phi(w_i^\top x_j) + \xi_j)^4])^{1/2} (\mathbb{E}[(x_j^\top \alpha)^8])^{1/2} (\mathbb{E}[\|x_j\|^4])^{1/2}$$
$$\lesssim (\|w_i\|^{p+1} + |\phi(0)| + \vartheta)^2 d.$$

Furthermore we have,

$$\max_{\|a\|=1} (\mathbb{E}[(a^\top C_i(x_j) a)^2])^{1/2} \lesssim (\mathbb{E}[(\phi(w_i^\top x_j) + \xi_j)^4])^{1/4} \lesssim \|w_i\|^{p+1} + |\phi(0)| + \vartheta.$$

Then by Claim E.6, with probability at least $1 - t$,

$$\left\| m_{4,i}(\overline{w}_i^\top \alpha)^2 \overline{w}_i \overline{w}_i^\top - \frac{1}{|S|} \sum_{x_j \in S} C_i(x_j) \right\|$$

$$\le \left\| m_{4,i}(\overline{w}_i^\top \alpha)^2 \overline{w}_i \overline{w}_i^\top - \mathbb{E}[C_i(x_j)] \right\| + \left\| \mathbb{E}[C_i(x_j)] - \frac{1}{|S|} \sum_{x_j \in S} C_i(x_j) \right\|$$

$$\lesssim \epsilon |m_{4,i}|(\overline{w}_i^\top \alpha)^2 + \epsilon.$$

Summing up all components $i \in [k]$, we proved Eq (9). Eq (7) and Eq (8) can be shown similarly.

$\square$

**Lemma B.18.** *Let $V \in \mathbb{R}^{d \times k}$ be an orthogonal matrix. Let $\widehat{R}_3$ be computed in Line 20 of Algorithm 7 and $R_3 = P_3(V, V, V)$. Suppose*

$$m_0 = \min_{i \in [k]}\{|m_{j_3,i}|^2 (\overline{w}_i^\top \alpha)^{2(j_3-3)}\}$$

*and*

$$|S| \gtrsim d \cdot \mathrm{poly}(\kappa, \vartheta, \log(d/t))/(\epsilon^2 m_0)$$

*then with probability at least $1 - t$,*

$$\|R_3 - \widehat{R}_3\| \lesssim \epsilon \sum_{i=1}^k |v_i m_{j_3,i} (\overline{w}_i^\top \alpha)^{j_3-3}| + \epsilon.$$

*Proof.* From the definition of $R_3$, it suffices to bound $\|M_3(V, V, V) - \widehat{M}_3(V, V, V)\|$ and $\|M_4(V, V, V, \alpha) - \widehat{M}_4(V, V, V, \alpha)\|$. The proof is similar to the previous one.

Specifically, we show that with probability at least $1 - t/4$,

$$\|M_3(V, V, V) - \widehat{M}_3(V, V, V)\| \lesssim \epsilon \sum_{i=1}^k |v_i m_{3,i}| + \epsilon. \tag{10}$$

$$\|M_4(V, V, V, \alpha) - \widehat{M}_4(V, V, V, \alpha)\| \lesssim \epsilon \sum_{i=1}^k |v_i m_{4,i}(\overline{w}_i^\top \alpha)| + \epsilon. \tag{11}$$

Recall that for sample $(x_j, y_j) \in S$, $y_j = \sum_{i=1}^k v_i \sigma(w_i^\top x_j) + \xi_j$ where $\xi_j$ is independent of $x_j$. Consider each component $i \in [k]$. Define $T_i(x_j), S_i(x_j) \in \mathbb{R}^{k \times k \times k}$:

$$
\begin{aligned}
T_i(x_j) &= (\sigma(w_i^\top x_j) + \xi_j) \\
&\quad \cdot \left(x_i^\top \alpha \cdot v(x)^{\otimes 3} - (V^\top \alpha)\widetilde{\otimes}(v(x) \otimes v(x)) - \alpha^\top x \cdot v(x)\widetilde{\otimes}I + (V^\top \alpha)\widetilde{\otimes}I\right), \\
S_i(x_j) &= \mathbb{1}(\|x_j\| \le B) \cdot T_i(x_j)
\end{aligned}
$$

where $v(x) = V^\top x$. Flatten $T_i(x_j)$ along the first dimension to obtain $B_i(x_j) \in \mathbb{R}^{k \times k^2}$, flatten $S_i(x_j)$ along the first dimension to obtain $C_i(x_j) \in \mathbb{R}^{k \times k^2}$.

From Claim E.7, $\mathbb{E}[B_i(x_j)] = m_{4,i}(\alpha^\top \overline{w}_i)(V^\top \overline{w}_i)\mathrm{vec}((V^\top \overline{w}_i)(V^\top \overline{w}_i)^\top)^\top$. Therefore we have,

$$\|\mathbb{E}[B_i(x)]\| = |m_{4,i}(\alpha^\top \overline{w}_i)| \cdot \|V^\top \overline{w}_i\|^3.$$

We calculate

$$
\begin{aligned}
\|\mathbb{E}_{\xi_j}[B_i(x_j)]\| \lesssim &(|w_i^\top x_j|^{p+1} + |\phi(0)|) \cdot ((x_j^\top \alpha)^2 \|V^\top x_j\|^3 \\
&+ 3\|V^\top x_j\|^3 + 3|x_j^\top \alpha|\|V^\top x_j\|\sqrt{k} + 3\|V^\top \alpha\|\sqrt{k}) \\
\lesssim &\sqrt{k} \cdot \|w_i\|^{p+1}\|x_j\|^{p+6}.
\end{aligned}
$$

By Assumption B.5, using Claim E.1 and $B \ge d \cdot \mathrm{poly}\log(d)$,

$$
\begin{aligned}
&\|\mathbb{E}[C_i(x_j)] - m_{4,i}(\alpha^\top \overline{w}_i)(V^\top \overline{w}_i)\mathrm{vec}((V^\top \overline{w}_i)(V^\top \overline{w}_i)^\top)^\top\| \\
&\lesssim \mathbb{E}[\mathbb{1}_{\|x_j\| \le B}\sqrt{k}\|w_i\|^{p+1}\|x_j\|^{p+6}] \\
&\le \epsilon.
\end{aligned}
$$

For any constant $t \in (0, 1)$, we have with probability $1 - t$,

$$
\begin{aligned}
\|C_i(x_j)\| \lesssim &(|w_i^\top x_j|^{p+1} + |\phi(0)| + |\xi_j|) \cdot ((x_j^\top \alpha)^2 \|V^\top x_j\|^3 \\
&+ 3\|V^\top x_j\|^3 + 3|x_j^\top \alpha|\|V^\top x_j\|\sqrt{k} + 3\|V^\top \alpha\|\sqrt{k}) \\
\lesssim &(\|w_i\|^{p+1} + |\phi(0)| + \vartheta)k^{3/2}\mathrm{poly}(\log(d/t))
\end{aligned}
$$

where the first step comes from Assumption B.5 and the second step comes from Claim E.2 and Claim E.3.

Using Claim E.4, we have

$$\left\|\mathbb{E}[C_i(x_j)C_i(x_j)^\top]\right\| \lesssim \left(\mathbb{E}\left[(\phi(w_i^\top x_j) + \xi_j)^4]\right)^{1/2} \left(\mathbb{E}\left[(\alpha^\top x_j)^4]\right)^{1/2} \left(\mathbb{E}\left[\|V^\top x_j\|^6]\right)^{1/2}$$
$$\lesssim (\|w_i\|^{p+1} + |\phi(0)| + \vartheta)^2 k^{3/2}.$$

and

$$\left\|\mathbb{E}[C_i(x_j)^\top C_i(x_j)]\right\|$$
$$\lesssim \left(\mathbb{E}[(\phi(w_i^\top x_j) + \xi_j)^4]\right)^{1/2} \left(\mathbb{E}[(\alpha^\top x_j)^4]\right)^{1/2} \left(\mathbb{E}[\|V^\top x_j\|^4]\right)^{1/2}$$
$$\cdot \left(\max_{\|A\|_F=1} \mathbb{E}\left[\langle A, (V^\top x_j)(V^\top x_j)^\top\rangle^4\right]\right)^{1/2}$$
$$\lesssim (\|w_i\|^{p+1} + |\phi(0)| + \vartheta)^2 k^2.$$

Furthermore we have,

$$\max_{\|a\|=\|b\|=1} \left(\mathbb{E}\left[(a^\top C_i(x_j)b)^2]\right)^{1/2}$$
$$\lesssim \left(\mathbb{E}[(\phi(w_i^\top x_j) + \xi_j)^4]\right)^{1/4} \left(\mathbb{E}\left[(\alpha^\top x_j)^4]\right)^{1/4} \max_{\|a\|=1} \left(\mathbb{E}\left[(a^\top V^\top x_j)^4]\right)^{1/2}$$
$$\cdot \max_{\|A\|_F=1} \left(\mathbb{E}\left[\langle A, (V^\top x_j)(V^\top x_j)^\top\rangle^4\right]\right)^{1/2}$$
$$\lesssim (\|w_i\|^{p+1} + |\phi(0)| + \vartheta)k.$$

Then by Claim E.6, with probability at least $1 - t$,

$$\left\|m_{4,i}(\alpha^\top \overline{w}_i)(V^\top \overline{w}_i)\text{vec}((V^\top \overline{w}_i)(V^\top \overline{w}_i)^\top)^\top - \frac{1}{|S|}\sum_{x_j \in S} C_i(x_j)\right\|$$
$$\leq \left\|m_{4,i}(\alpha^\top \overline{w}_i)(V^\top \overline{w}_i)\text{vec}((V^\top \overline{w}_i)(V^\top \overline{w}_i)^\top)^\top - \mathbb{E}[C_i(x_j)]\right\|$$
$$+ \left\|\mathbb{E}[C_i(x_j)] - \frac{1}{|S|}\sum_{x_j \in S} C_i(x_j)\right\|$$
$$\lesssim \epsilon|v_i m_{4,i}(\overline{w}_i^\top \alpha)| + \epsilon.$$

Summing up all neurons $i \in [k]$, we proved Eq (11). Eq (10) can be shown similarly.

$\square$

**Lemma B.19.** *Let $\widehat{Q}_1$ and $\widehat{Q}_2$ be computed in Line 28 of Algorithm 7. Let $Q_1$ be defined by Eq 4 and $Q_2$ be defined by Eq 5. Suppose*

$$m_0 = \min_{i \in [k]}\{|m_{j_1,i}|^2(\overline{w}_i^\top \alpha)^{2(j_1-1)}, |m_{j_2,i}|^2(\overline{w}_i^\top \alpha)^{2(j_2-2)}\}$$

*and*

$$|S| \gtrsim d \cdot \text{poly}(\kappa, \vartheta, \log(d/t))/(\epsilon^2 m_0)$$

*then with probability at least $1 - t$,*

$$\|Q_1 - \widehat{Q}_1\| \lesssim \epsilon \sum_{i=1}^{k} |v_i m_{j_1,i}(\overline{w}_i^\top \alpha)^{j_1-1}| + \epsilon,$$

$$\|Q_2 - \widehat{Q}_2\| \lesssim \epsilon \sum_{i=1}^{k} |v_i m_{j_2,i}(\overline{w}_i^\top \alpha)^{j_2-2}| + \epsilon.$$

*Proof.* Recall the expression of $Q_1$ and $Q_2$,

$$Q_1 = M_{l_1}(I, \underbrace{\alpha, \cdots, \alpha}_{(j_1-1) \ \alpha\text{'s}}) = \sum_{i=1}^{k} v_i c_{j_1} \|w_i\|^{p+1} (\alpha^\top \overline{w}_i)^{j_1-1} \overline{w}_i,$$

$$Q_2 = M_{j_2}(V, V, \underbrace{\alpha, \cdots, \alpha}_{(j_2-2) \ \alpha\text{'s}}) = \sum_{i=1}^{k} v_i c_{j_2} \|w_i\|^{p+1} (\alpha^\top \overline{w}_i)^{j_2-2} (V^\top \overline{w}_i)(V^\top \overline{w}_i)^\top.$$

The proof is essentially similar to Lemma B.17 and Lemma B.18. $\qquad\square$

We also use the following Lemmata from [47, 90].

**Lemma B.20** (Adapted from Theorem 3 of [47]). *Given a tensor $\widehat{T} = \sum_{i=1}^{k} \pi_i u_i^{\otimes 3} + \epsilon R \in \mathbb{R}^{d \times d \times d}$. Assume incoherence $u_i^\top u_j \leq \mu$. Let $L_0 := (\frac{50}{1-\mu^2})^2$ and $L \geq L_0 \log(15d(k-1)/t)^2$. Then there exists an algorithm such that, with probability at least $1 - t$, for every $u_i$, the algorithm returns a $\widetilde{u}_i$ such that*

$$\|\widetilde{u}_i - u_i\|_2 \leq O\left( \frac{\sqrt{\|\pi\|_1 \pi_{\max}}}{\pi_{\min}^2} \cdot \frac{\|V^\top\|_2^2}{1-\mu^2} \cdot (1 + C(t)) \right) \epsilon + o(\epsilon),$$

*where $C(t) := \log(kd/t)\sqrt{d/L}$ and $V$ is the inverse of the full-rank extension of $(u_1 \ldots u_k)$ with unit-norm columns.*

**Lemma B.21** (Adapted from Lemma E.6 of [90]). *Let $P_2$ be defined as in Definition B.16 and $\widehat{P}_2$ be its empirical version calculated in Line 4 of Algorithm 7. Let $U \in \mathbb{R}^{d \times k}$ be the orthogonal column span of $W \in \mathbb{R}^{d \times k}$. Assume $\|\widehat{P}_2 - P_2\| \leq s_k(P_2)/10$. Let $C$ be a large enough positive number such that $C > 2\|P_2\|$. Then after $T = O(\log(1/\epsilon))$ iterations, the $V \in \mathbb{R}^{d \times k}$ computed in Algorithm 7 will satisfy*

$$\|UU^\top - VV^\top\| \lesssim \|\widehat{P}_2 - P_2\|/s_k(P_2) + \epsilon,$$

*which implies*

$$\|(I - VV^\top)w_i\| \lesssim (\|\widehat{P}_2 - P_2\|/s_k(P_2) + \epsilon)\|w_i\|.$$

**Lemma B.22** (Adapted from Lemma E.13 in [90]). *Let $U \in \mathbb{R}^{d \times k}$ be the orthogonal column span of $W^*$. Let $V \in \mathbb{R}^{d \times k}$ denote an orthogonal matrix satisfying that $\|VV^\top - UU^\top\| \leq \widehat{\delta}_2 \lesssim 1/(\kappa^2\sqrt{k})$. For each $i \in [k]$, let $\widehat{u}_i$ denote the vector satisfying $\|s_i\widehat{u}_i - V^\top \overline{w}_i\| \leq \widehat{\delta}_3 \lesssim 1/(\kappa^2\sqrt{k})$. Let $Q_1$ be defined as in Eq (4) and $\widehat{Q}_1$ be the empirical version of $Q_1$ such that $\|Q_1 - \widehat{Q}_1\| \leq \widehat{\delta}_4\|Q_1\| \leq \frac{1}{4}\|Q_1\|$.*

*Let $z^* \in \mathbb{R}^k$ and $\widehat{z} \in \mathbb{R}^k$ be defined as in Eq (6) and Line 29. Then*

$$|\widehat{z}_i - z_i^*| \leq (\kappa^4 k^{3/2}(\widehat{\delta}_2 + \widehat{\delta}_3) + \kappa^2 k^{1/2}\widehat{\delta}_4)\|z^*\|_1.$$

**Lemma B.23** (Adapted from Lemma E.14 in [90]). *Let $U \in \mathbb{R}^{d \times k}$ be the orthogonal column span of $W^*$ and $V$ be an orthogonal matrix satisfying that $\|VV^\top - UU^\top\| \leq \widehat{\delta}_2 \lesssim 1/(\kappa\sqrt{k})$. For each $i \in [k]$, let $\widehat{u}_i$ denote the vector satisfying $\|s_i\widehat{u}_i - V^\top \overline{w}_i^*\| \leq \widehat{\delta}_3 \lesssim 1/(\sqrt{k}\kappa^3)$.*

*Let $Q_2$ be defined as in Eq (5) and $\widehat{Q}_2$ be the empirical version of $Q_2$ such that $\|Q_2 - \widehat{Q}_2\|_F \leq \widehat{\delta}_4\|Q_2\|_F \leq \frac{1}{4}\|Q_2\|_F$. Let $r^* \in \mathbb{R}^k$ and $\widehat{r} \in \mathbb{R}^k$ be defined as in Eq (6) and Line 29. Then*

$$|\widehat{r}_i - r_i^*| \leq (k^3\kappa^8\widehat{\delta}_3 + \kappa^2 k^2\widehat{\delta}_4)\|r^*\|.$$

Now we are in the position of proving Theorem B.13.

*Proof.* Consider Algorithm 7. First, by Lemma B.21 and Lemma B.17, we have

$$\begin{aligned}
\|VV^\top \overline{w}_i - \overline{w}_i\| &\leq (\|\widehat{P}_2 - P_2\|/s_k(P_2) + \epsilon) \\
&\leq (\text{poly}(k, \kappa)\|\widehat{P}_2 - P_2\| + \epsilon) \\
&\leq \text{poly}(k, \kappa)\epsilon.
\end{aligned} \tag{12}$$

Next, combining Lemma B.20 and Lemma B.18, we have

$$\|V^\top \overline{w}_i - s_i \widehat{u}_i\| \leq \text{poly}(k, \kappa) \|\widehat{R}_3 - R_3\| \leq \epsilon \text{poly}(k, \kappa). \tag{13}$$

It thus follows that

$$
\begin{aligned}
\|\overline{w}_i - s_i V \widehat{u}_i\| &\leq \|VV^\top \overline{w}_i - \overline{w}_i\| + \|VV^\top \overline{w}_i - V s_i \widehat{u}_i\| \\
&= \|VV^\top \overline{w}_i - \overline{w}_i\| + \|V^\top \overline{w}_i - s_i \widehat{u}_i\| \\
&\leq \epsilon \text{poly}(k, \kappa),
\end{aligned} \tag{14}
$$

where the first step applies triangle inequality and the last step uses Eq (12) and Eq (13).

We proceed to bound the error in $\widehat{r}$ and $\widehat{z}$. We have,

$$
\begin{aligned}
|\widehat{r}_i - r_i^*| &\lesssim \text{poly}(k, \kappa)(\|Q_2 - \widehat{Q}_2\| + \|s_i \widehat{u}_i - V^\top \overline{w}_i\|) \cdot \|r^*\| \\
&\lesssim \epsilon \text{poly}(k, \kappa) \cdot \|r^*\|
\end{aligned} \tag{15}
$$

where the first step comes from Lemma B.23 and the second step comes from Lemma B.19 and Eq (14). Furthermore,

$$
\begin{aligned}
|\widehat{z}_i - z_i^*| &\lesssim \text{poly}(k, \kappa) \cdot (\|Q_1 - \widehat{Q}_1\| + \|s_i \widehat{u}_i - V^\top \overline{w}_i\| + \|VV^\top - UU^\top\|) \cdot \|z^*\|_1 \\
&\lesssim \epsilon \text{poly}(k, \kappa) \|z^*\|_1,
\end{aligned} \tag{16}
$$

where the first step comes from Lemma B.22 and the second step comes from combining Lemma B.19, Lemma B.21, and Eq (14). Finally, combining Eq (15), Eq (16) and Eq (14), the output in Line 32 satisfies $\|\widehat{w}_i - w_i\|_F \leq \epsilon \text{poly}(k, \kappa) \cdot \|w_i\|_F$. Since $v_i$ are discrete values, they are exactly recovered. $\qquad\square$

### B.5.2 Exact recovery of neural networks from noiseless data

In this section we prove Theorem B.15. Similar to Appendix B.5.1, denote $W = [w_1, \cdots, w_k]^\top$ where $w_i \in \mathbb{R}^d$ and $\widehat{W} = [\widehat{w}_1, \cdots, \widehat{w}_k]^\top$. We use $\mathcal{D}$ to denote the distribution of $x \sim \mathcal{N}(0, I_d)$ and $y = \langle v, \sigma(Wx) \rangle$. We define the empirical loss for explored features and the population loss as follows,

$$L_n(\widehat{W}) = \frac{1}{2n} \sum_{(x,y) \in S^{(1)}} \left( \sum_{i=1}^{k} v_i \sigma(\widehat{w}_i^\top x_i) - y_i \right)^2, \tag{17}$$

$$L(\widehat{W}) = \frac{1}{2} \mathbb{E}_{\mathcal{D}} \left[ \left( \sum_{i=1}^{k} v_i \sigma(\widehat{w}_i^\top x) - y \right)^2 \right]. \tag{18}$$

---

**Algorithm 8** Using method of moments and gradient descent to recover neural network parameters

---

1: **procedure** NEURALNETRECOVERY($S = \{(x_i, y_i) : i \in [n]\}$)
2:      Let $S^{(1)} \leftarrow \{(x, y) \in S : \|x\| \leq B\}$
3:      Compute $(v, \widehat{W}) \leftarrow$ NEURALNETNOISYRECOVERY($S^{(1)}$)
4:      Find $W^{(1)}$ as the global minimum of $L_n(\cdot)$, where $L_n(\cdot)$ is defined in Eq (17).
5:      **Return** $(v, W^{(1)})$
6: **end procedure**

---

**Definition B.24.** Let $s_i$ be the $i$-th singular value of $W$, $\lambda := \prod_{i=1}^{k}(s_i/s_k)$. Let $\tau = (3s_1/2)^{4p} / \min_{z \in [s_k/2, 3s_1/2]} \{\rho^2(z)\}$.

We use the follow results adapted from [90]. The only difference is that the rewards are potentially truncated if $\|x\| \geq B$, and due to $B = d \cdot \text{poly} \log(d)$ we can bound its difference between standard Gaussian in the same way as Appendix B.5.1.

**Lemma B.25** (Concentration, adapted from Lemma D.11 in [90]). *Let samples size $n \geq \epsilon^{-2}d\tau\text{poly}(\log(d/t))$, then with probability at least $1 - t$,*

$$\|\nabla^2 L_n(W) - \nabla^2 L(W)\| \lesssim ks_1^{2p}\epsilon + \text{poly}(B_W, d)e^{-\Omega(d)}.$$

**Lemma B.26** (Adapted from Lemma D.16 in [90]). *Assume activation $\sigma(\cdot)$ satisfies Assumption B.8 and Assumption B.7. Then for any $t \in (0, 1)$, if $n \geq d \cdot \text{poly}(\log(d/t))$, with probability at least $1 - t$, for any $\widehat{W}$ (which is not necessarily to be independent of samples) satisfying $\|W - \widehat{W}\| \leq s_k/4$, we have*

$$\|\nabla^2 L_n(\widehat{W}) - \nabla^2 L_n(W)\| \leq ks_1^p\|W - \widehat{W}\|d^{(p+1)/2}.$$

Now we prove Theorem B.15.

*Proof.* The exact recovery consists of first finding (exact) $v$ and (approximate) $\widehat{W}$ close enough to $W$ by tensor method (Appendix B.5.1), and then minimizing the empirical loss $L_n(\cdot)$. We will prove that $L_n(\cdot)$ is locally strongly convex, thus we find the precise $W$.

From Lemma D.3 from [90] we know:

$$\Omega(\rho(s_k)/\lambda)I \preceq \nabla^2 L(W) \preceq O(ks_1^{2p})I. \tag{19}$$

Combining Lemma B.25, $d \geq \log(B_W/\lambda)$, and $n \geq \frac{k^2\lambda^2s_1^{4p}}{\rho^2(s_k)}d\tau\text{poly}(\log(d/t))$, we know $\nabla^2 L_n(W)$ must be positive definite.

Next we uniformly bound Lipschitzness of $\nabla^2 L_n$. From Lemma B.26 there exists a universal constant $c$, such that for all $\widehat{W}$ that satisfies $\|W - \widehat{W}\| \leq cks_1^{2p}/(ks_1^pd^{(p+1)/2}) = cs_1^pd^{-(p+1)/2}$, $\nabla L_n^2(\widehat{W}) \gtrsim ks_1^{2p}$ holds uniformly. So there is a unique miminizer of $L_n$ in this region.

Notice $L_n(W) = 0$, therefore we can find $W$ by directly minimizing the empirical loss as long as we find any $\widehat{W}$ in this region. This can be achieved by tensor method in Appendix B.5.1. We thus complete the proof. $\qquad\square$

## C   Omitted Proofs in Section 4

For the proofs of Theorem 4.2, Example 4.3, and Example 4.4, we refer the readers to [36].

**Lemma C.1.** *Consider the polynomial family $\mathcal{F}_\mathcal{V}$ of dimension $D$. Assume that $n > 2D$. For any $E \in \mathbb{R}^d$ that is of positive measure, by sampling $n$ samples $\{x_i\}$ i.i.d. from $\mathbb{P}_{x \in \mathcal{N}(0, I_d)}(\cdot|x \in E)$ and observing the noiseless feedbacks $y_i = f^*(x_i)$, one can almost surely uniquely determine the $f^*$ by solving the system of equations $y_i = f(x_i)$, $i = 1, \ldots, n$, for $f \in \mathcal{F}_\mathcal{V}$.*

*Proof.* By Theorem 4.2, there exists a set $N \in \mathbb{R}^d \times \ldots \mathbb{R}^d$ of Lebesgue measure zero, such that if $(x_1, \cdots, x_n) \notin N$, one can uniquely determine the $f^*$ by the observations on the $n$ samples. Therefore, we only need to show that with probability 1, the sampling procedure returns $(x_1, \ldots, x_n) \notin N$. This is because

$$
\begin{aligned}
\mathbb{P}(x_1, \ldots, x_n \in N) &= \mathbb{P}_{x_i \in \mathcal{N}(0, I_d)}((x_1, \ldots, x_n) \in N \mid x_1, \ldots, x_n \in E) \\
&= \frac{\mathbb{P}_{x_i \in \mathcal{N}(0, I_d)}((x_1, \ldots, x_n) \in N \cap (E \times \cdots \times E))}{\mathbb{P}_{x_i \in \mathcal{N}(0, I_d)}((x_1, \ldots, x_n) \in (E \times \cdots \times E))} \\
&= \frac{0}{[\mathbb{P}_{x_1 \in \mathcal{N}(0, I_d)}(x_1 \in E)]^n} \\
&= 0.
\end{aligned}
$$

$\qquad\square$

By Lemma C.1 above, it is not hard to see that Algorithms 4 and 5 work.

# D Omitted Constructions and Proofs in Subsection 4.1

**Construction of the Reward Functions**  The following construction of the polynomial hard case is adopted from [36].

Let $d$ be the dimension of the feature space. Let $e_i$ denotes the $i$-th standard orthonormal basis of $\mathbb{R}^d$, i.e., $e_i$ has only one 1 at the $i$-th entry and 0's for other entries. Let $p$ denote the highest order of the polynomial. We assume $d \gg p$. We use $\Lambda$ to denote a subset of the $p$-th multi-indices

$$\Lambda = \{(\alpha_1, \ldots, \alpha_p) | 1 \leq \alpha_1 \leq \cdots \leq \alpha_p \leq d\}.$$

For an $\alpha = (\alpha_1, \ldots, \alpha_p) \in \Lambda$, denote $M_\alpha = e_{\alpha_1} \otimes \cdots \otimes e_{\alpha_p}$, $x_\alpha = e_{\alpha_1} + \cdots + e_{\alpha_p}$.

The model space $\mathcal{M}$ is a subset of rank-1 $p$-th order tensors, which is defined as $\mathcal{M} = \{M_\alpha | \alpha \in \Lambda\}$. We define two subsets of feature space $\mathcal{F}_0$ and $\mathcal{F}$ as $\mathcal{F}_0 = \{x_\alpha | \alpha \in \Lambda\}$, $\mathcal{F} = \text{conv}(\mathcal{F}_0)$. For $M_\alpha \in \mathcal{M}$, $x \in \mathcal{F}$, define $r(M_\alpha, x)$ as $r(M_\alpha, x) = \langle M_\alpha, x^{\otimes p} \rangle = \prod_{i=1}^{p} \langle e_{\alpha_i}, x \rangle$. We assume that for each level $h$, there is a $M^{(h)} = M_{\alpha^{(h)}} \in \mathcal{M}$, and the noiseless reward is $r_h(s, a) = r(M^{(h)}, \phi_h(s, a))$.

We have the following properties.

**Proposition D.1** ([36]). *For $M_\alpha \in \mathcal{M}$ and $x_{\alpha'} \in \mathcal{F}_0$, we have*

$$r(M_\alpha, x_{\alpha'}) = \mathbb{I}_{\{\alpha = \alpha'\}}.$$

**Proposition D.2.** *For $M_\alpha \in \mathcal{M}$, we have*

$$\max_{x \in \mathcal{F}} r(M_\alpha, x) = 1.$$

*proof of Proposition D.2.* For all $x \in \mathcal{F}$, since $\mathcal{F} = \text{conv}(\mathcal{F}_0)$, we can write

$$x = \sum_{\alpha \in \Lambda} p_\alpha(e_{\alpha_1} + \cdots + e_{\alpha_p}),$$

where $\sum_{\alpha \in \Lambda} p_\alpha = 1$ and $p_\alpha \geq 0$. Therefore,

$$r(M_{\alpha'}, x) = \prod_{i=1}^{p} \langle e_{\alpha'_i}, x \rangle.$$

Plug in the expression of $x$, we have

$$\langle e_{\alpha'_i}, x \rangle = \sum_\alpha p_\alpha \langle e_{\alpha'_i}, e_{\alpha_1} + \cdots + e_{\alpha_p} \rangle$$
$$= \sum_\alpha p_\alpha \mathbb{I}_{\{\alpha'_i \in \alpha\}}$$
$$\leq \sum_\alpha p_\alpha = 1.$$

Therefore,

$$r(M_{\alpha'}, x) = \prod_{i=1}^{p} \langle e_{\alpha'_i}, x \rangle$$
$$= \left( \sum_\alpha p_\alpha \mathbb{I}_{\{e_{\alpha'_1} \in \alpha\}} \right) \cdots \left( \sum_\alpha p_\alpha \mathbb{I}_{\{e_{\alpha'_p} \in \alpha\}} \right)$$
$$\leq 1.$$

Finally, since $r(M_{\alpha'}, x_{\alpha'}) = 1$, we have $\max_{x \in \mathcal{F}} r(M_\alpha, x) = 1$. $\qquad\square$

**MDP constructions**  Consider a family of MDPs with only two states $\mathcal{S} = \{S_{\text{good}}, S_{\text{bad}}\}$. The action set $\mathcal{A}$ is set to be $\mathcal{F}$. Let $f$ be a mapping from $\mathcal{F}$ to $\mathcal{F}_0$ such that $f$ is identity when restricted to $\mathcal{F}_0$. For all level $h \in [H]$, we define the feature map $\phi_h : S \times A \to \mathcal{F}$ to be

$$\phi_h(s, a) = \begin{cases} a & \text{if } s = S_{\text{good}}, \\ f(a) & \text{if } s = S_{\text{bad}}. \end{cases}$$

Given an unknown sequence of indices $\alpha^{(1)}, \ldots, \alpha^{(H)}$, the reward function at level $h$ is $r_h(s, a) = r(M_{\alpha^{(h)}}, \phi_h(s, a))$. Specifically, we have

$$r_h(S_{\text{good}}, a) = r(M_{\alpha^{(h)}}, a), \ r_h(S_{\text{bad}}, a) = r(M_{\alpha^{(h)}}, f(a)).$$

The transition $P_h$ is constructed as

$$P_h(S_{\text{bad}}|s, a) = 1 \text{ for all } s \in \mathcal{S}, a \in \mathcal{A}.$$

This construction means it is impossible for the online scenarios to reach the good state for $h > 1$.

The next proposition shows that $Q_h^*$ is polynomial realizable and falls into the case of Example 4.4.

**Proposition D.3.** *We have for all $h \in [H]$ and $s \in \mathcal{S}$, $a \in \mathcal{A}$, $V_h^*(s) = H - h + 1$ and $Q_h^*(s, a) = r_h(s, a) + H - h + 1$. Furthermore, $Q_h^*(s, a)$, viewed as the function of $\phi_h(s, a)$, is a polynomial of the form $q_h(U_h \phi_h(s, a))$ for some degree-$p$ polynomial $q_h$ and $U_h \in \mathbb{R}^{p \times d}$.*

*proof of Proposition D.3.* First notice that by Proposition D.2, for all $h \in [H]$ and $s \in \mathcal{S}$, we have

$$\max_{a \in \mathcal{A}} r_h(s, a) = 1.$$

Therefore, by induction, suppose we have proved for all $s'$, $V_{h+1}^*(s') = H - h$, then we have

$$\begin{aligned}
V_h^*(s) &= \max_{a \in \mathcal{A}} Q_h^*(s, a) \\
&= \max_{a \in \mathcal{A}} \{r_h(s, a) + \mathbb{E}_{s' \sim P_h(\cdot|s,a)}[V_{h+1}^*(s')]\} \\
&= 1 + H - h.
\end{aligned}$$

Then we have $Q_h^*(s, a) = r_h(s, a) + H - h + 1$.

Furthermore, we have

$$\begin{aligned}
Q_h^*(s, a) &= r_h(s, a) + H - h + 1 \\
&= r(M_{\alpha^{(h)}}, \phi_h(s, a)) + H - h + 1 \\
&= \prod_{i=1}^{p} \langle e_{\alpha_i^{(h)}}, \phi_h(s, a) \rangle + H - h + 1 \\
&= q_h(U_h \phi_h(s, a)),
\end{aligned}$$

where $q_h(x_1, \ldots, x_p) = x_1 x_2 \cdots x_p + (H - h + 1)$ and $U_h \in \mathbb{R}^{p \times d}$ is a matrix with $e_{\alpha_i^{(h)}}$ as the $i$-th row. $\qquad \square$

**Theorem D.4.** *Under the online RL setting, any algorithm needs to play at least $(\binom{d}{p} - 1) = \Omega(d^p)$ episodes to identify $\alpha^{(2)}, \ldots, \alpha^{(H)}$ and thus to identify the optimal policy.*

*proof of Theorem D.4.* Under the online RL setting, any algorithm enters and remains in $S_{\text{bad}}$ for $h > 1$. When $s_h = S_{\text{bad}}$, no matter what $a_h$ the algorithm chooses, we have $\phi_h(s_h, a_h) = f(a_h) \in \mathcal{F}_0$. Notice that for any $M_{\alpha^{(h)}} \in \mathcal{M}$ and any $x_\alpha \in \mathcal{F}_0$, we have $r(M_{\alpha^{(h)}}, x_\alpha) = \mathbb{I}_{\{\alpha = \alpha^{(h)}\}}$ as Proposition D.1 suggests. Hence, we need to play $(\binom{d}{p} - 1)$ times at level $h$ in the worst case to find out $\alpha^{(h)}$. The argument holds for all $h = 2, 3, \ldots, H$. $\qquad \square$

**Theorem D.5.** *Under the generative model setting, by querying $2d(p + 1)^p H = O(dH)$ samples, we can almost surely identify $\alpha^{(1)}, \alpha^{(2)}, \ldots, \alpha^{(H)}$ and thus identify the optimal policy.*

*proof of Theorem D.5.* By Proposition D.3, we know that $Q_h^*(s, a)$, viewed as the function of $\phi_h(s, a)$, falls into the case of Example 4.4 with $k = p$.

Next, notice that for all $h \in [H]$, $\{\phi_h(s, a) \mid s \in \mathcal{S}, a \in \mathcal{A}\} = \mathcal{F}$. Although $\mathcal{F}$ is not of positive measure, we can actually know the value of $Q_h^*$ when $\phi_h(s, a)$ is in $\text{conv}(\mathcal{F}, \mathbf{0})$ since the reward is $p$-homogenous. Specifically, for every feature of the form $c \cdot \phi_h(s, a)$, where $0 \leq c \leq 1$ and $\phi_h(s, a) \in \mathcal{F}$, the reward is $c^p$ times the reward of $(s, a)$. Therefore, to get the reward at $c \cdot \phi_h(s, a)$, we only need to query the generative model at $(s, a)$ of level $h$, and then multiply the reward by $c^p$.

Notice that $\text{conv}(\mathcal{F}, \mathbf{0})$ is of positive Lebesgue measure. By Theorem 4.7, we know that only $2d(p + 1)^p H = O(dH)$ samples are needed to determine the optimal policy almost surely.

$\qquad \square$

# E   Technical claims

**Claim E.1.** Let $\chi^2(d)$ denote $\chi^2$-distribution with degree of freedom $d$. For any $t > 0$ we have,

$$\Pr_{z \sim \chi^2(d)} (z \geq d + 2t + 2\sqrt{dt}) \leq e^{-t}$$

We use the following facts from [90].

**Claim E.2.** Given a fixed vector $z \in \mathbb{R}^d$, for any $C \geq 1$ and $n \geq 1$, we have

$$\Pr_{x \sim \mathcal{N}(0,I_d)}[|\langle x, z \rangle|^2 \leq 5C\|z\|^2 \log n] \geq 1 - 1/(nd^C).$$

**Claim E.3.** For any $C \geq 1$ and $n \geq 1$, we have

$$\Pr_{x \sim \mathcal{N}(0,I_d)}[\|x\|^2 \leq 5Cd \log n] \geq 1 - 1/(nd^C).$$

**Claim E.4.** Let $a, b, c \geq 0$ be three constants, let $u, v, w \in \mathbb{R}^d$ be three vectors, we have

$$\mathbb{E}_{x \sim \mathcal{N}(0,I_d)}\left[|u^\top x|^a|v^\top x|^b|w^\top x|^c\right] \approx \|u\|^a\|v\|^b\|w\|^c.$$

**Claim E.5.** Let $M_j, j \in [4]$ be defined in Definition B.11. For each $j \in [4]$, $M_j = \sum_{i=1}^k v_i m_{j,i} \overline{w}_i^{\otimes j}$.

**Claim E.6.** Let $\mathcal{B}$ denote a distribution over $\mathbb{R}^{d_1 \times d_2}$. Let $d = d_1 + d_2$. Let $B_1, B_2, \cdots B_n$ be i.i.d. random matrices sampled from $\mathcal{B}$. Let $\overline{B} = \mathbb{E}_{B \sim \mathcal{B}}[B]$ and $\widehat{B} = \frac{1}{n}\sum_{i=1}^n B_i$. For parameters $m \geq 0, \gamma \in (0,1), \nu > 0, L > 0$, if the distribution $\mathcal{B}$ satisfies the following four properties,

$$(1) \qquad \Pr_{B \sim \mathcal{B}}[\|B\| \leq m] \geq 1 - \gamma;$$

$$(2) \qquad \left\|\mathbb{E}_{B \sim \mathcal{B}}[B]\right\| > 0;$$

$$(3) \qquad \max\left(\left\|\mathbb{E}_{B \sim \mathcal{B}}[BB^\top]\right\|, \left\|\mathbb{E}_{B \sim \mathcal{B}}[B^\top B]\right\|\right) \leq \nu;$$

$$(4) \qquad \max_{\|a\|=\|b\|=1}\left(\mathbb{E}_{B \sim \mathcal{B}}\left[\left(a^\top Bb\right)^2\right]\right)^{1/2} \leq L.$$

Then we have for any $0 < \epsilon < 1$ and $t \geq 1$, if

$$n \geq (18t \log d) \cdot (\nu + \|\overline{B}\|^2 + m\|\overline{B}\|\epsilon)/(\epsilon^2\|\overline{B}\|^2) \quad \text{and} \quad \gamma \leq (\epsilon\|\overline{B}\|/(2L))^2$$

with probability at least $1 - 1/d^{2t} - n\gamma$,

$$\|\widehat{B} - \overline{B}\| \leq \epsilon\|\overline{B}\|.$$

**Claim E.7.** Let $P_2$ and $P_3$ be defined in Definition B.16. Then

$$P_2 = \sum_{i=1}^k v_i m_{j_2,i}(\alpha^\top \overline{w}_i)^{j_2-2}\overline{w}_i^{\otimes 2}$$

and

$$P_3 = \sum_{i=1}^k v_i m_{j_3,i}(\alpha^\top \overline{w}_i)^{j_3-3}\overline{w}_i^{\otimes 3}.$$