# OpenReview forum: "Going Beyond Linear RL: Sample Efficient Neural Function Approximation"
_NeurIPS.cc/2021/Conference — NeurIPS 2021 Poster_

### Official Review · Reviewer_92NY · 2021-07-12

**Rating:** 6
**Confidence:** 4

**Summary:**

This paper studies sample efficient RL with two-layer neural network function approximation with a generative model. In particular, they prove that they can learn an optimal policy with poly samples with only realizability in deterministic MDP, and that the sample complexity for learning a near-optimal policy in stochastic MDPs is in an order of $\epsilon^{-2}$ in two cases: policy completeness assumption and Bellman completeness assumption. The algorithms they use are based on simple (non-optimistic) value regression.

**Ethical Concerns:**

No concern.

**Limitations And Societal Impact:**

The authors have partly addressed the limitations of their work and fully addressed the societal impact of their work.

**Main Review:**

**Strong points**:
- Sample efficient RL with deep neural network function approximation is an important and timely topic
- The sample complexity with neural network function approximation does not suffer from the curse of dimensionality  (i.e., sample efficiency is possible in RL with neural network function approximation)

**Weak points**:
- The main guarantees are direct results of the lemmas of the exact network recovery from noisy/noiseless data [Zhong et al, 2017]. What is a technical innovation in this paper?
- I think learning with a simulator (a generative model) is rather not a practical setting as in practice, data is sequentially obtained from the beginning of an episode. Can the result with neural network function approximation here be extended to such online setting? Currently, the paper has the result in an online setting with polynomial realizability.
- The paper claims that “this is the first paper that shows sample efficient RL is possible with neural net function approximation”. It does not seem so. For example, Yang et al., 2020 “On Function Approximation in Reinforcement Learning: Optimism in the Face of Large State Spaces” already obtains a sample efficient RL in online setting with neural function approximation using optimism. Without optimism, Pan Xu and Quanquan Gu “A Finite-Time Analysis of Q-Learning with Neural Network Function Approximation” obtains sample efficiency of Q-learning with neural network function approximation in online setting with Markovian data.

**Minor comments**
- The abstract only mentions neural network function approximation but in the main paper, it also considers polynomial function.
- Line 139: typo, \kappa = s_{\max} / s_{\min}
- Line 93: Missing literature, I think you should consider adding Jin et al, 2021, “Bellman Eluder Dimension: New Rich Classes of RL Problems, and Sample-Efficient Algorithms”, as you discuss sample efficient RL with function approximation.

**Questions for authors**:
- What is a rationale for the sample efficiency of simple value regression without optimism in your proposed algorithms Alg 1,2,3? What is the key to such a simple algorithm that can lead to the obtained sample efficiency? Probably due to the generative model that we can obtain good coverage of the state-action space that does not require intricate exploration such as optimism, but I would like to hear the authors discuss more on this.
- More discussion on how mild are the policy-complete and Bellman-complete assumptions.

------------------ POST-REBUTTAL----------------------

Thank you the authors for providing a detailed response to my concerns. After reading the authors' responses and other reviews, I am happy to raise my initial score to 6 for the following reasons: the paper has made a solid contribution in showing the feasibility of sample efficient RL with non-linear neural network function approximation in the generative setting. Technically, though the paper relies heaviely on the parameter recovery theorem in non-linear neural networks and an extension to RL setting in the generative model is quite natural and straightforward, the result in the paper does complement the literature of RL with non-linear function approximation and can be helpful to the community. My initial score was due to mainly two issues: the technical contribution of the paper and the result's significance. Though there is no significant technical contribution as in my initial review, I misinterpreted the paper result when comparing it with the literature RL with overparameterized neural networks. I encourage the authors to take the discussion into the revision.



**Time Spent Reviewing:**

5

---

> ### Author Response · Authors · 2021-08-10
> **Our paper should be evaluated on what it proves, not how it proved the results.**
>
> Thanks for reading our results and thanks for your valuable comments. We respond to the questions and comments in the following.
>
> 1. "What is a technical innovation in this paper?"
> This paper should be evaluated on what it proves, not how it proved the results. The technical tools are all standard, and we readily cited them throughout. The point here is that with standard tools, we can show the most general function approximation in simulator RL, by reducing simulator RL to parameter recovery in supervised learning. This allows us to transfer any model that allows parameter recovery (e.g neural networks) to the simulator RL setting.
>
> Though our techniques are simple, they displace many conventional wisdoms in the current RL theory beliefs. As an example, the literature suggests that function approximation with neural networks is impossible (e.g. https://arxiv.org/abs/2102.04168 or https://arxiv.org/abs/2104.06970 which prove the eluder dimension/ sample complexity of learning a single relu is exp(d) even in the bandit setting). Our results show that in the generative RL, in fact even two-layer networks are learnable. In other words, the current best result for parameter recovery in  neural networks can be automatically plugged into our proof and show RL is no harder than learning a neural network. In other words, the results of this paper can be viewed as reducing RL in the generative setting to parameter learning. As simple as this observation is, it is incredibly powerful and does give the most general function approximation results in generative RL.
>
> Again, the results of a theoretical paper should be evaluated on its results, not the complexity of its proofs /arguments.
>
>
>
>
> Our main contribution is this simple reduction and then identifying interesting models that allow for parameter recovery and then allow efficient RL. We achieve this by the following two techniques: 1) analysis of dynamic programming algorithms when $L_\infty$ recovery guarantee is possible; 2) provable parameter recovery of two-layer neural networks and Admissible Polynomial Families. In the first technique, we consider the following settings: realizable, realizable + Bellman complete, realizable + policy complete, which is the first unified theoretical treatment as far as we know. Using this framework, other spectral methods (e.g. [Janzamin et al. 2015, Ge et al. 2017, Fornasier et al. 2021]) for neural network recovery would also enjoy provable sample efficiencies. In the second technique, we establish the first recovery guarantee of Admissible Polynomial Families and generalize the spectral method of [Zhong et al. 2017] from noiseless samples to noisy samples.
>
> 2. "Can the result with neural network function approximation here be extended to such online setting?"
>
>     - We thank the reviewer for raising this interesting future direction. We believe that our results can be extended to online settings by assuming the features of each time step has positive measure over a norm $d \log^{O(1)} d$ ball. Under this assumption, we can apply explore and then commit algorithm and achieve $O(\mathrm{poly} (d) T^{⅔})$​​​​​ regret. It is a major open problem to attain poly(d) T^{1/2} regret under neural net function approximation.
>
> 3. "What is a rationale for the sample efficiency of simple value regression without optimism in your proposed algorithms Alg 1,2,3? What is the key to such a simple algorithm that can lead to the obtained sample efficiency?"
>
>     - Since the algorithm works under generative models, it only needs to guarantee good exploration. The key ingredient is that in the specified settings (realizable + deterministic transition, realizable + policy completeness, realizable + Bellman completeness), the exploration can be decoupled into optimal planning via dynamic programming and model recovery. The latter can be achieved using spectral methods from Gaussian explorations.
>
> 4. "More discussion on how mild are the policy-complete and Bellman-complete assumptions."
>
>     Our comments on the assumptions are as follows. We will add detailed discussions in the revision.
>
>     - Realizability is a *necessary* assumption in studying general function approximation RL, which means the function class contains optimal action-value function. When this fails to hold, one needs to pay some misspecification error in the worst case. This assumption is *standard* and commonly used in function approximation literature.
>     - Completeness is another *standard* assumption widely used in function approximation [Wang et al. 2020, Jin et al. 2020, Jin et al. 2021]. Bellman completeness says that the function class contains the image of the Bellman operator. It is *necessary* in value based algorithms utilizing Bellman update because if the Bellman update is not included in the function class, it will incur a misspecification error that propagates exponentially through time steps. In fact, it is known that even in linear function approximation absence of completeness will cause exponential sample complexity lower bounds [Du et al. 2020]. Policy completeness is another *standard* assumption [Lagoudakis et al. 2003, Lattimore et al. 2020]. This assumption is commonly use in least-squares policy improvement (LSPI) algorithms. Notice that linear MDPs [Jin et al. 2020] satisfy both Bellman completeness and Policy completeness.
>
> 5. We thank the reviewer for pointing out typos and missing literature. We will improve the writing in revision.
>
> 6. "The paper claims that “this is the first paper that shows sample efficient RL is possible with neural net function approximation”. It does not seem so. For example, Yang et al., 2020 “On Function Approximation in Reinforcement Learning: Optimism in the Face of Large State Spaces” already obtains a sample efficient RL in online setting with neural function approximation using optimism. Without optimism, Pan Xu and Quanquan Gu “A Finite-Time Analysis of Q-Learning with Neural Network Function Approximation” obtains sample efficiency of Q-learning with neural network function approximation in online setting with Markovian data."
>
>      - Thank you for pointing out this imprecise statement. The major point is that previous work all studies the Neural Tangent Kernel function approximation and this paper is the first beyond this scope. Kernel methods and linear RL are widely studied, but ours is the first that does not linearize the network and treat it as a kernel method. As shown in the cited papers in the related work, kernel methods suffer from a d^p sample complexity of learning a rank 1 simple degree function such as x_1^p, but a neural net only needs d samples. The major gain here is in sample complexity from considering neural nets, rather than linearized networks which are kernel methods. We will correct this statement in the final version.
>
> Refs:
>
> [Janzamin et al. 2015]. Majid Janzamin, Hanie Sedghi, and Anankumar Anima. Beating the perils of nonconvexity: Guaranteed training of neural networks using tensor methods. arXiv preprint arXiv:1506.08473, 2015.
>
> [Ge et al. 2017]. Rong Ge, Jason D. Lee, and Tengyu Ma. Learning one-hidden-layer neural networks with landscape design. arXiv preprint arXiv:1711.00501, 2017.
>
> [Fornasier et al. 2021]. Fornasier, M., Klock, T. & Rauchensteiner, M. Robust and Resource-Efficient Identification of Two Hidden Layer Neural Networks. Constr Approx (2021).
>
> [Wang et al. 2020] Wang, R., Salakhutdinov, R., and Yang, L. F. Provably efficient reinforcement learning with general value function approximation. arXiv preprint arXiv:2005.10804. 2020.
>
> [Jin et al. 2020] Jin, C., Yang, Z., Wang, Z., and Jordan, M. I. Provably efficient reinforcement learning with linear function approximation. In Conference on Learning Theory, pp. 2137–2143. PMLR, 2020.
>
> [Jin et al. 2021] Jin, C., Liu, Q., and Miryoosefi, S. Bellman eluder dimension: New rich classes of rl problems, and sample-efficient algorithms. arXiv preprint arXiv:2102.00875, 2021.
>
> [Du et al. 2020] Du, S. S., Kakade, S. M., Wang, R., and Yang, L. F., Is a good representation sufficient for sample efficient reinforcement learning? In International Conference on Learning Representations, 2020.
>
> [Lattimore et al. 2020] Lattimore, T. and Szepesvari, C. Learning with good feature representations in bandits and in rl with a generative model. In International Conference on Machine Learning (ICML), 2020.
>
> [Lagoudakis et al. 2003] Lagoudakis, M. G. and Parr, R. Least-squares policy iteration. Journal of machine learning research, 4(Dec): 1107–1149, 2003.

---

> > ### Comment · Reviewer_92NY · 2021-08-24
> > **Response to the author's comment**
> >
> > I appreciate the authors for providing very detailed comments on my concerns. In my original review, I misinterpreted the significance of the paper results in the literature of sample efficient RL with overparameterized neural networks. After reading the other reviews, the author's response, and https://arxiv.org/pdf/2102.04168.pdf, though the technical innovation in the paper is quite simple and standard, I think the contribution of the paper is solid, that is sample efficient RL with **non-linear** function approximation is possible in a generative setting. Thus, I am happy to increase my score to 6 with the condition that the authors discuss and compare clearly their results for non-linear function approximation with the other line of work using overparameterized neural nets in RL  (Yang et al., 2020 “On Function Approximation in Reinforcement Learning: Optimism in the Face of Large State Spaces” and Pan Xu and Quanquan Gu “A Finite-Time Analysis of Q-Learning with Neural Network Function Approximation”). For example, the problem differences, the key challenges in each problem, and why these two approaches are orthogonal (?) could be discussed.

---

> > > ### Author Response · Authors · 2021-08-25
> > > **Neural tangent kernel does not include actual neural nets.**
> > >
> > >
> > > Thanks for acknowledging this work. Please find our response below:
> > > - Even for a single ReLU neuron, the RKHS norm is exponential [Yehudai et al. 2020]. Thus neural tangent kernel does *not* contain two-layer neural networks. In fact, the paragraph of ‘Eluder dimension’ in the related work also applies to NTK and any kernel method with low information gain [Huang et al. 2021].
> > > - To compare, this paper does not assume overparameterization and thus the model is non-linear. Additionally, the model presented in this paper is not captured by Eluder dimension or Bellman Eluder dimension, while the overparameterized neural networks are RKHS models that are known as special cases of low Eluder dimension. Furthermore, [Dong et al. 2021] shows that neural networks have large Eluder dimension, therefore a long line of work that studies general function approximation, such as Bellman rank, Bilinear rank, Eluder dimension, and Bellman Eluder dimension can not be easily applied. This is the main challenge in deep RL theory.
> > >
> > > Refs:
> > >
> > > [Yehudai et al. 2020] On the Power and Limitations of Random Features for Understanding Neural Networks. Gilad Yehudai et al. 2020.
> > >
> > > [Huang et al. 2021] A short note on the relationship of eluder dimension and information gain. Kaixuan Huang. 2021.

---

### Official Review · Reviewer_3peY · 2021-07-15

**Rating:** 7
**Confidence:** 4

**Summary:**

The paper considers RL with neural function approximation, extending significantly previous works that only focuses on linear feature approximation. In particular, the authors propose (1) an efficient algorithm using two-layer NNs in a generative model assuming completeness, and (2) an efficient algorithm with two-layer NNs but only assuming realizability.

The results significantly extends prior research on linear methods, or methods with bounded eluder dimension.

**Limitations And Societal Impact:**

Limitations and scopes of the theoretical results are clearly stated. The strong assumptions required are reemphasized in the conclusion and directions for future research are proposed.

No apparent negative societal impact.

**Main Review:**

- Originality: the paper is one of the few papers that consider reinforcement learning in (the more general) neural network approximation regime. Related works are clearly cited and ~the work has made significant contributions different from existing research~. Please see the updated comment. I realized I neglected the result's reliance on [1] and have adjusted my score accordingly.

- Quality: the theorems and results in the paper are well justified.

- Clarity: proofs are clear and well-written.

- Significance: being one of the first papers studying neural network approximation, the paper could pave way for future research in this area. ~The results are technically sound and significant.~ See updated comment.

[1] Zhong, Kai, et al. "Recovery guarantees for one-hidden-layer neural networks." International conference on machine learning. PMLR, 2017

**Time Spent Reviewing:**

8

---

> ### Author Response · Authors · 2021-08-10
> **Thanks for your positive feedback!**
>
> Thanks so much for reading and understanding our results, and thanks for your valuable comments!

---

> ### Comment · Reviewer_3peY · 2021-08-19
> **Update w.r.t. [Zhong et al. 2017]**
>
> Having read other reviewers' comments, I realized I have missed how the paper's results rely on [Zhong et al. 2017]. Given this reliance, I would like to lower my score to a 5.
>
> The major contributions of the paper lies in directly analyzing an RL algorithm using the neural function approximation class itself, rather than NTKs or other linear models as a surrogate. Unfortunately, a lot of the analysis utilizes existing recovery guarantees for one-layer neural networks.
>
> While it is true that a theory paper should be evaluated based on the results rather than the proof's complexity, such a reliance on existing results does limit the originality of the paper. The strong realizability assumption does limit the scope of contribution, and I am not convinced by the author's response to reviewer qeUq's first point. While [4, 5] both uses strong realizability assumptions, these papers also provide stronger, provably efficient algorithms, as well as results under misspecification.

---

> > ### Author Response · Authors · 2021-08-20
> > **Response to the update.**
> >
> > Please find the response below.
> >
> > 1. Reliance on [Zhong et al. 2017].
> >    - As we have mentioned in responses, although our techniques are simple, they displace many conventional wisdoms in the current RL theory beliefs. For example, the literature suggests that function approximation with neural networks is impossible (e.g. [Malik et al. 2021], [Dong et al. 2021] or [Li et al. 2021] which prove the eluder dimension/ sample complexity of learning a single relu is exp(d) even in the bandit setting). Our results show that in the generative RL, in fact even two-layer networks are learnable. In other words, the current best result for parameter recovery in neural networks can be automatically plugged into our proof and show RL is no harder than learning a neural network. In other words, the results of this paper can be viewed as reducing RL in the generative setting to parameter learning. As simple as this observation is, it is incredibly powerful and does give the most general function approximation results in generative RL.
> >    - In fact, other spectral methods (e.g. [Janzamin et al. 2015, Ge et al. 2017, Fornasier et al. 2021]) for neural network recovery would also work in our framework. We do not claim novelty in any of these as they are all well-known to experts or trivial. Nor do we attempt to obfuscate how the theorems are proved, as we adequately discussed how technique details come from [Zhong et al. 2017] in the proof.
> > 2. Realizability assumption.
> >    - Realizability is a common and standard assumption that is widely used in RL function approximation. In [Jin et al. 2020], the regret becomes linear wrt T without this assumption. In addition, [Wang et al. 2020] considers the misspecification model of low Bellman error and this model still satisfies the realizability assumption. Furthermore, we consider neural network function approximation which has stronger expressiveness and larger capacity. Thus it seems unfair to compare our results with theirs in terms of realizability assumption.
> >
> >
> >
> >
> >
> > Refs:
> >
> > [Dong et al. 2021] Kefan Dong, Jiaqi Yang, and Tengyu Ma. Provable model-based nonlinear bandit and reinforcement learning: Shelve optimism, embrace virtual curvature. arXiv preprint arXiv:2102.04168, 2021.
> >
> > [Li et al. 2021] Gene Li, Pritish Kamath, Dylan J. Foster, and Nathan Srebro. Eluder Dimension and Generalized Rank. arXiv preprint arXiv:2104.06970, 2021.
> >
> > [Malik et al. 2021] Dhruv Malik, Aldo Pacchiano, Vishwak Srinivasan, and Yuanzhi Li. Sample Efficient Reinforcement Learning In Continuous State Spaces: A Perspective Beyond Linearity. arXiv preprint arXiv:2106.07814, 2021.
> >
> > [Janzamin et al. 2015]. Majid Janzamin, Hanie Sedghi, and Anankumar Anima. Beating the perils of nonconvexity: Guaranteed training of neural networks using tensor methods. arXiv preprint arXiv:1506.08473, 2015.
> >
> > [Ge et al. 2017]. Rong Ge, Jason D. Lee, and Tengyu Ma. Learning one-hidden-layer neural networks with landscape design. arXiv preprint arXiv:1711.00501, 2017.
> >
> > [Fornasier et al. 2021]. Fornasier, M., Klock, T. & Rauchensteiner, M. Robust and Resource-Efficient Identification of Two Hidden Layer Neural Networks. Constr Approx (2021).
> >
> > [Wang et al. 2020] Wang, R., Salakhutdinov, R., and Yang, L. F. Provably efficient reinforcement learning with general value function approximation. arXiv preprint arXiv:2005.10804. 2020.
> >
> > [Jin et al. 2020] Jin, C., Yang, Z., Wang, Z., and Jordan, M. I. Provably efficient reinforcement learning with linear function approximation. In Conference on Learning Theory, pp. 2137–2143. PMLR, 2020.

---

> > > ### Comment · Reviewer_3peY · 2021-08-20
> > > **Response to the authors’ comment**
> > >
> > > I would like to thank the authors sincerely for constantly engaging in the conversation and the thorough responses.
> > >
> > > Sorry for flip-flopping constantly, but, in my defense, it is difficult to assess the contributions of the paper given that it is a **novel** combination of **existing** research and, depending on the reviewers’ perspective and personal preferences, will lead to incredibly different scores for the same paper. The setting is novel, the theory is sound, and the results are promising, yet the proof depend on other existing research. Depending on reviewers’ different perspectives on what constitutes good research, the review for the paper can range from critical or incredibly favorable.
> > >
> > > Quick summary of the lengthy comment: after carefully rereading the papers, reviewers’ comments, the authors’ responses, and the references, I am now confident in my original assessment that the paper should be accepted. While the initial score of 9 may be too high, the paper is at least a 6 given that it applied existing deep learning results in an interesting new field.
> > >
> > > Below I summarize what I gleaned from the discussion phase as well as rereads of the paper and related literature.
> > >
> > > Pros of the paper:
> > > 1. MDP + Neural Function Approximation: directly tacking MDP + neural FA, rather than using NTKs as a linear surrogate, has yet to be analyzed. The paper is novel in the sense that it answers when does such a approach work. Given the wide popularity and success of such methods in practice, and the relatively pessimistic outlook on the approach in theory, working to resolve such a dichotomy should be highly appreciated.
> > > 2. RE: Eluder dimension: the authors pointed out that the difficulty of RL with neural FA under the simulation model, especially the recent line of work focusing on eluder dimension and hardness of RL. In this respect, by reducing RL with simulator to exact recovery of a neural network, the author have already responded to a popular line of current research and identified a regime under which eluder dimension may not be the best characterization of learnability. I do believe that the related works section on Eluder Dimension should be updated to *emphasize more* on this fact. For instance, the author’s response to reviewer 92NY highlights such a contribution, and would be greatly appreciated if it is also included in the main body of the text.
> > > 3. Implications on the hardness of RL with function approximation in the simulator setting: the paper provides a followup to an existing line of research on the hardness of RL with simulator models (kindly see the papers the authors mentioned in the paper). However, this line of research often lies in the tabular MDP regime and, to my best knowledge, the paper is the first one to consider the problem in neural FA regime. Whether the generative model regime is interesting may be debatable, yet the paper does provide an interesting new take at an existing problem.
> > >
> > > As an aside, I do believe the authors are doing a better job highlighting the paper’s contributions in the discussion phase and that has led me to identify some contributions not easily identifiable from the paper itself. If these contributions were highlighted in the related works section as well as the introduction section, the paper could have a better reception among reviewers.
> > >
> > > Limitations of the paper:
> > > 1. Reliance on [Zhong et al. 2017] and technical contributions of the paper
> > > 2. Realizability assumption
> > > 3. Computation (see the critique on efficient algorithms for solving the system of equations)
> > >
> > > The limitations have been thoroughly discussed elsewhere in the discussion threads, so will not waste anyone’s time reiterating these points. However, I do believe in light of the pros of the paper, these limitations could be forgiven (see below)
> > > - If we view the paper as reducing the hardness of RL with generative models to exact recovery of NNs, it is more than justifiable to rely on existing literature in exact recovery. The authors also improve upon existing exact recovery research by allowing for noise, which remains a contribution, albeit the significance of the contribution may be debatable. Regardless, if we view the **reduction** as the more important insight, then the reliance on [Zhong et al. 2017] is a moot point, given that the reduction, rather than the ensuing theoretical analysis, is the meat of the results.
> > >     - In the same vein, the strong realizability assumption is also justified. However, other papers that use this analysis have at least some misspecification results, and similar results in the paper could definitely better complete the argument.
> > >     - Since computation efficiency is not usually a major point of discussion in theoretical ML research, I do think that even without an efficient algorithm, the results are strong and interesting enough.
> > >
> > > *I do believe there is some confusion w.r.t. the main contributions of the paper*. I believe at the current stage, the paper is less of a general sample efficient approach to solve MDPs with neural FA, but more of an answer to the line of works posed a recent line of works on eluder dimensions and hardness of RL with NNs. The first task would probably require more realistic assumptions w.r.t. completeness or realizability, and those are fair if the goal of the paper were to analyze RL with neural FA in general. However, the paper does identify regimes where neural FAs work, showed how to make neural FAs work. Furthermore, even in the online regime without simulators, the paper answers the question of “when does NNs with poly activation work?” (See the generic feature assumption). Admittedly the proof techniques are not the strong suit of the paper, but to me, that is slightly besides the point, as the more significant contribution lies in identifying when RL + neural FA works. (To authors and other reviewers: **PLEASE CORRECT ME IF THIS PARAGRAPH DOESN’T DO YOUR PERSPECTIVE ENOUGH JUSTICE**).
> > >
> > > After reconsidering the pros and cons, as well as checking the NeurIPS reviewer guideline, I do decide that the paper should be accepted. It is true that the assumptions are strong and the analysis depends significantly on existing works. However, at least according to the reviewer guideline, the novel connection the authors make between RL and exact recovery of NNs should be rewarded, and is a major contribution in its own right (“a novel combination of well-known techniques”). Despite the lack of exciting new insight or novel analysis methods for RL with NNs, the paper does answer a series of question on hardness of RL. The paper have, at the very least, shown a regime under which RL using function approximations with high eluder dimension is efficient, and should be accepted. It could definitely use some revisions, better highlighting some of its contributions that the authors mentioned in the comments and I do believe such revisions would make the paper much more appealing and better sell the paper. However, I do believe the novel connection made by the authors are interesting and the paper should be accepted.
> > >
> > > I hope the authors and the area chair can see how this paper can lead to different opinions from the different reviewers (and, if you will kindly let me be the first one to poke fun at myself, the paper can lead to different opinions from the same reviewer who is a bit too quick to pull the trigger). In light of the reviewer guideline, however, I now stand by the opinion that the paper should be accepted for its novel connection, as I do personally believe that such novel insights are just as important as novel theoretical techniques.

---

> > > > ### Author Response · Authors · 2021-08-25
> > > > **Response to the updates.**
> > > >
> > > > Thank you for acknowledging this paper and pointing out how it is different from using neural tangent kernels in RL. We agree that 'the more significant contribution lies in identifying when RL + neural FA works'. To complete the argument, we comment that our algorithm also works in non-realizable settings, with an additional polynomial factor of misspecification error in the suboptimality gap. It is known that the method of moments is robust to misspecification errors (e.g. section 5 of [Janzamin et al. 2015]). Besides, approximation value iteration decouples the suboptimality gap into the errors of learning Q-function in each time step. Therefore the misspecification errors add linearly into the suboptimality gap, and the guarantee would become as follows: it takes $O(\mathrm{poly}(d)/\epsilon^2)$ trajectories to learn a policy $\pi$ such that $Q^{\pi^\ast}(s_1) - Q^{\pi}(s_1) \leq \epsilon + \mathrm{poly}(H \zeta)$ where $\zeta$ is the $l_\infty$ misspecification error (which might come from non-completeness as well). Thus the algorithm works well when the misspecification error is very small (which is often the case in neural network function approximations).

---

### Official Review · Reviewer_AzY2 · 2021-07-16

**Rating:** 3
**Confidence:** 5

**Summary:**

This paper focuses on analyzing sampling efficiency of nonlinear reinforcement learning with neural network approximations of the $Q$ function. It reduces the sampling complexity from $O(d^{\text{poly}(1/\epsilon)})$ to $O(\text{poly}(d) \exp(k))$ where $k$ is the width of the hidden layer of a two-layer fully connected neural network. The idea is described accurately and the paper is overall well written.

**Limitations And Societal Impact:**

Yes.

**Main Review:**

The paper considers both neural network and polynomial approximations of the $Q$ function. Since polynomial approximation does not scale well with the dimensionality, the analysis on the sampling complexity of neural network-based $Q$ function approximation is more interesting. In the neural network setting, the RL algorithms, i.e., Algorithm 1, 2 and 3 are quite standard RL algorithms, so the novelty does not lie in algorithm development. In my opinion, the main contributions of this paper should be the proof of Theorem 4.4, 4.5 and 4.6. However, the main idea of the proofs are not included in the main paper.

When digging into Appendix, it becomes apparent that the key improvement of the sampling complexity are based on the results in

Kai Zhong, Zhao Song, Prateek Jain, Peter L. Bartlett, and Inderjit S. Dhillon. Recovery guarantees for one-hidden-layer neural networks. Proceedings of the Thirty-fourth International Conference on Machine Learning (ICML), 70, 2017.

For example, the proof of Theorem 4.4 is given in Appendix as Theorem A.1. In the proof of Theorem A.1, the key step is the use of Theorem A.14. When looking at Theorem A.14, the proof is provided in Appendix A.4.2 from Line 715 to Line 725. The main ingredient of the ten-line proof is the used of Lemma D.3, Lemma D.11 and Lemma D.16 in [Zhong et al. 2017]. Similar strategy is used to prove Theorem 4.5 and 4.6. Therefore, I think Theorem 4.4 - 4.6 are just simple extension of the results in [Zhong et al. 2017] to the RL setting, so the theoretical contribution of this paper is not significant.

On the other hand, I strongly disagree with the way the theoretical results are presented, because I don't think the importance of [Zhong et al. 2017] is explicitly and sufficiently acknowledged in the main text. In fact, the paper [Zhong et al. 2017] is only cited twice in the main text. Even though it is mentioned in Line 49 that [Zhong et al. 2017] is one of the main technique used in this paper, but it is not completely not clear that the complexity of NN-based approximation in Table 1 comes from the neural network recovery result in [Zhong et al. 2017].

Additionally, there is a lack of justification of the appropriateness of Assumption 3.1 and 3.2. It looks like the assumptions were made on purpose so that the results in [Zhong et al. 2017] can be used. It would be helpful if the authors can provide some RL examples that satisfy the assumptions.



**Time Spent Reviewing:**

5

---

> ### Author Response · Authors · 2021-08-10
> **the results of a theoretical paper should be evaluated on its results, not the complexity of its proofs /arguments.**
>
> Thanks for reading our results and write this review. We will improve the presentation in the revision as you suggested. We ask that you reconsider your score after reading our response which clarifies our position.
>
>  As we stated clearly in the proof sketches of Section 4 under theorem 4.4 and 4.5, the techniques of this paper are straightforward: a) Value iteration works under linfty errors, b) Parameter recovery + Lipschitz implies linfty control, and c) Spectral/tensor methods enable parameter recovery for a wide variety of models including neural networks.  We do not claim novelty in any of these  as they are all well-known to experts or trivial. Nor do we attempt to obfuscate how the theorems are proved.
>
> On the other hand, recent work in RL suggests that function approximation with neural networks is impossible (e.g. https://arxiv.org/abs/2102.04168 or https://arxiv.org/abs/2104.06970 which suggest the eluder dimension/ sample complexity of learning a single relu is exp(d) even in the bandit setting). Our results show that in the generative RL, in fact even two-layer networks are learnable. In other words, the current best result for two-layer networks can be automatically plugged into our proof and show RL is no harder than learning a neural network. In other words, the results of this paper can be viewed as reducing RL in the generative setting to parameter learning. As simple as this observation is, it is incredibly powerful and does give the most general function approximation results in generative RL. The results of a theoretical paper should be evaluated on its results, not the complexity of its proofs /arguments.
>
> 1. We would like to comment that the main contribution of this paper is identifying interesting models that allow for parameter recovery and then allow efficient RL. In fact, other spectral methods (e.g. [Janzamin et al. 2015, Ge et al. 2017, Fornasier et al. 2021]) for neural network recovery would also work in our framework. Therefore, generalizing the spectral method of [Zhong et al. 2017] from noiseless samples to noisy samples is *not* the main contribution of this paper. We have fully acknowledge [Zhong et al. 2017] in the appendix and pointed out how its techniques help in our setting. We will add more discussions in the main text.
> 2. Our comments on the assumptions are as follows. We will add detailed discussions in the revision.
>
>     - The presented assumptions on neural networks are regularity conditions that cover a wide class of neural networks. It is shown in [Zhong et al. 2017] that ReLU, leaky ReLU, and squared ReLU activations all satisfy these assumptions.
>
>     - Realizability is a *necessary* assumption in studying general function approximation RL, which means the function class contains the optimal action-value function. When this fails to hold, one needs to pay some misspecification error in the worst case. This assumption is *standard* and commonly used in function approximation literature.
>     - Completeness is another *standard* assumption widely used in function approximation [Wang et al. 2020, Jin et al. 2020, Jin et al. 2021]. Bellman completeness says that the function class contains the image of the Bellman operator. It is *necessary* in value-based algorithms utilizing Bellman update because if the Bellman update is not included in the function class, it will incur a misspecification error that propagates exponentially through time steps. In fact, it is known that even in linear function approximation absence of completeness will cause exponential sample complexity lower bounds [Du et al. 2020]. Policy completeness is another *standard* assumption [Lagoudakis et al. 2003, Lattimore et al. 2020]. This assumption is commonly use in least-squares policy improvement (LSPI) algorithms. Notice that linear MDPs [Jin et al. 2020] satisfy both Bellman completeness and Policy completeness.
>
>
> Refs:
>
> [Janzamin et al. 2015]. Majid Janzamin, Hanie Sedghi, and Anankumar Anima. Beating the perils of nonconvexity: Guaranteed training of neural networks using tensor methods. arXiv preprint arXiv:1506.08473, 2015.
>
> [Ge et al. 2017]. Rong Ge, Jason D. Lee, and Tengyu Ma. Learning one-hidden-layer neural networks with landscape design. arXiv preprint arXiv:1711.00501, 2017.
>
> [Fornasier et al. 2021]. Fornasier, M., Klock, T. & Rauchensteiner, M. Robust and Resource-Efficient Identification of Two Hidden Layer Neural Networks. Constr Approx (2021).
>
> [Wang et al. 2020] Wang, R., Salakhutdinov, R., and Yang, L. F. Provably efficient reinforcement learning with general value function approximation. arXiv preprint arXiv:2005.10804. 2020.
>
> [Jin et al. 2020] Jin, C., Yang, Z., Wang, Z., and Jordan, M. I. Provably efficient reinforcement learning with linear function approximation. In Conference on Learning Theory, pp. 2137–2143. PMLR, 2020.
>
> [Jin et al. 2021] Jin, C., Liu, Q., and Miryoosefi, S. Bellman eluder dimension: New rich classes of rl problems, and sample-efficient algorithms. arXiv preprint arXiv:2102.00875, 2021.
>
> [Du et al. 2020] Du, S. S., Kakade, S. M., Wang, R., and Yang, L. F., Is a good representation sufficient for sample efficient reinforcement learning? In International Conference on Learning Representations, 2020.
>
> [Lattimore et al. 2020] Lattimore, T. and Szepesvari, C. Learning with good feature representations in bandits and in rl with a generative
> model. In International Conference on Machine Learning (ICML), 2020.
>
> [Lagoudakis et al. 2003] Lagoudakis, M. G. and Parr, R. Least-squares policy iteration. Journal of machine learning research, 4(Dec):
> 1107–1149, 2003.

---

> > ### Comment · Reviewer_AzY2 · 2021-08-23
> > **Additional comments on the theoretical contribution of this work**
> >
> > I really appreciate the authors' response to my comments. I also carefully read other reviewers' comments. My understanding about the authors' claim on the theoretical contribution is the proof of the following statement:
> >
> > "(S1): If the optimal policy is in the neural network family considered in [Zhong et. al, 2017], then the recovery results in [Zhong et. al 2017] can be directly extended to the RL setting without significant modification."
> >
> > Moreover, the authors also claimed in the response that the paper [Zhong et. al, 2017] in the above statement can be replaced by [Janzamin et al. 2015, Ge et al. 2017, Fornasier et al. 2021] or maybe any other paper on recovery of neural networks or any high-dimensional approximation schemes. However, I think the authors missed a big question behind it, i.e.,
> >
> > (Q1): For what kind of RL problems does the Realizability assumption (i.e., the optimal policy is in the neural network family) hold?
> >
> > Once this question is answered, we will be able to know the applicability of the recovery result. Moreover, I believe the polynomial policy and neural network policy (or any other nonlinear models in [Janzamin et al. 2015, Ge et al. 2017, Fornasier et al. 2021]) are applicable to different sets of RL problems. Thus, theoretical analysis on the applicability of neural network policies (answering Q1) should be the key topic for this work.
> >
> > Taking a step back, if the extension of the results in [Zhong et. al, 2017], i.e., proving the statement (S1), was very challenging, which requires novel and deep analysis techniques to accomplish the proof, this paper would deserve to be accepted, and (Q1) could be answered in a followup paper. This was the reason why I dug through the entire analysis including Appendix trying to find some novel analysis during the first round review. Unfortunately, it is not the case for this paper, so I think theoretical analysis on (Q1) needs to be done for both neural network and polynomial policies. Please correct me if I underestimate the contribution of this paper.

---

> > > ### Author Response · Authors · 2021-08-25
> > > **Response to the additional comments.**
> > >
> > > We would like to clarify that this paper does not assume the optimal policy is in the neural network family. The realizability assumption actually says that the optimal Q-function lies in the neural network family. We use this assumption because neural function approximations of Q-functions are widely used in practical RL, e.g. DQN, DDPG, rainbow, etc. In these works, people believe that neural networks can express the Q-functions and are essential to learn and generalize in large state spaces. Thus this assumption is made as a modeling assumption to understand the why and how deep rl scales and generalizes to large state spaces. Realizable Q function is the weakest assumption in deep rl theory ([Yang et al. 2020, Xu et al. 2020, Cai et al. 2020, Liu et al 2019, Wang et al. 2019]).
> > >
> > > To complete the picture, we also comment that our algorithm also works in non-realizable settings, with an additional polynomial factor of misspecification error in the suboptimality gap. It is known that the method of moments is robust to misspecification errors, e.g. section 5 of [Janzamin et al. 2015]. Besides, approximation value iteration decouples the suboptimality gap into the errors of learning Q-function in each time step. Therefore the misspecification errors add linearly into the suboptimality gap, and the guarantee would become as follows: it takes $O(\mathrm{poly}(d)/\epsilon^2)$ trajectories to learn a policy $\pi$ such that $Q^{\pi^\ast}(s_1) - Q^{\pi}(s_1) \leq \epsilon + \mathrm{poly}(H \zeta)$ where $\zeta$ is the $l_\infty$ misspecification error. Thus the algorithm works well when the misspecification error is very small (which is often the case in neural network function approximations).
> > >
> > > Refs:
> > >
> > > [Yang et al. 2020] On Function Approximation in Reinforcement Learning: Optimism in the Face of Large State Spaces. Zhuoran Yang et al. 2020.
> > >
> > > [Xu et al. 2020] A Finite-Time Analysis of Q-Learning with Neural Network Function Approximation. Pan Xu et al. 2020.
> > >
> > > [Cai et al. 2020] Neural Temporal-Difference and Q-Learning Provably Converge to Global Optima. Qi Cai et al. 2020.
> > >
> > > [Zhou et al. 2019] Neural contextual bandits with upper confidence bound-based exploration. Dongruo Zhou et al. 2019.
> > >
> > > [Liu et al. 2019] Neural proximal/trust region policy optimization attains globally optimal policy. Boyi Liu et al. 2019.
> > >
> > > [Wang et al. 2019] Neural Policy Gradient Methods: Global Optimality and Rates of Convergence. Lingxiao Wang et al. 2019.

---

> > > > ### Comment · Reviewer_AzY2 · 2021-08-30
> > > > **About the realizability assumption**
> > > >
> > > > I appreciate the authors' response to my comments. I apologize for being not very careful about the terminology I used in my comments. What I really meant in (S1) is the same as the authors' clarification. In fact, a more general statement of (S1) is that "If the target function is in the neural network family, ...", where the "target function" is the Q-function in this paper. However, I still think the authors' response relies on empirical conclusion, i.e.,
> > > >
> > > > "We use this assumption because neural function approximations of Q-functions are widely used in practical RL, e.g. DQN, DDPG, rainbow, etc. In these works, people believe that neural networks can express the Q-functions and are essential to learn and generalize in large state spaces."
> > > >
> > > > For a paper that claims novel theoretical contributions, using such empirical conclusion as the major assumption is not reasonable, because the goal of theoretical development is to convert empirical observations/results to rigorously proved theorems. Again, for the RL problem considered in this paper, there are two well-known empirical observations in practice, e.g., (1) NN is a good approximation scheme for the Q-function, and (2) the NN-based Q-function approximator can be well trained with reasonable complexity. The goal of this paper should be converting the two empirical observations into rigorously proved theorems. Again, if (2) is very challenging to prove, then this paper would deserve to be accepted. However, due to the assumption on realizability, proving (2) becomes an easy extension of the results in [Zhong et al. 2017]. In other words, the realizability assumption contains the actual challenges of the theoretical problem, so it should be proved in this paper, instead of using empirical observations. Therefore, I still maintain my score.

---

> > > > > ### Author Response · Authors · 2021-08-30
> > > > > **We do show (2) and realizability CANNOT BE PROVED**
> > > > >
> > > > > Dear Area chair, PC and  other Reviewers,
> > > > > Realizability is not an assumption that can be proved! This is like asking to prove that the optimal regressor f* is a neural net, this is an assumption. The entire field of statistical learning works with assumptions like this or chooses to compare to the best neural net (agnostic/best in class).
> > > > > We have already shown to you that our results also attain agnostic results also. Further we showed (2) by giving a polynomial-time algorithm.
> > > > >
> > > > > At this point, I can assure all the reviewers and Area chairs that we have completely addressed this reviewers concerns regarding assumptions (realizability and agnostic results are both proved) and computational (the algorithm is polytime)

---

### Official Review · Reviewer_qeUq · 2021-07-16

**Rating:** 6
**Confidence:** 3

**Summary:**

This paper studies reinforcement learning with a 1-hidden layer neural network and low-rank polynomial function approximation schemes.

Under mostly standard assumptions about realizability, completness etc, they provide algorithms mainly focusing on sample complexity in cases when there is access to a perfect generative model of the transitions and in the fully online setting, where planning is not possible in episodic reinforcement learning.

They improve upon the sample complexity with respect to existing baselines for the respective function approximation schemes.

**Limitations And Societal Impact:**

See above for limitations.

**Main Review:**

My main questions and concerns are as follows:

1)The baseline considered in Du et al 2020c, considers a weaker form of realizability assumption, where they can find the best function approximation for $Q^*$ in the given class upto a error $\delta > 0$. Now, since the class here are 1-layer neural networks or low-rank polynomials, there exists simple deterministic transitions that have complex Q functions (i.e, when using shallow networks this requires exponential width), (see Dong et al. 2020). Is it possible to comment on the tradeoff due to expressivity errors?

2) Could the authors clarify on the dependence on H, more explicitly if possible?

3) Theorem 5.2 states that there is a unique solution to the system of equations. Is there an efficient algorithm for it? Since it is used in Algorithms 4 and 5.

4) A large portion of the results in the neural network recovery is from Zhong et al. 2017, as a clarification is assumption A11 satisfied by ReLu activation?

Additional Reference:
Dong, Kefan, et al. "On the expressivity of neural networks for deep reinforcement learning." International Conference on Machine Learning. PMLR, 2020.

After Rebuttal:

 I raise my score after the authors made the necessary clarifications.

**Time Spent Reviewing:**

9

---

> ### Author Response · Authors · 2021-08-10
> **Thanks for your feedback!**
>
> Thanks for reading our results and thanks for your valuable comments. We respond to the questions and comments in the following. We ask that the reviewer consider increasing the score, given that we have addressed the comments and questions.
>
> 1. "The baseline considered in Du et al 2020c, considers a weaker form of realizability assumption, where they can find the best function approximation for Q∗ in the given class up to a error δ>0. Now, since the class here are 1-layer neural networks or low-rank polynomials, there exists simple deterministic transitions that have complex Q functions (i.e, when using shallow networks this requires exponential width), (see Dong et al. 2020). Is it possible to comment on the tradeoff due to expressivity errors?"
>
>     - We thank the reviewer for raising this interesting future direction of model misspecifications with neural network function approximation, and we agree that it takes exponential width to approximate Q function in the worst cases. However, neural function approximation works well in practical RL [1,2,3] since it has great expressiveness power *in general*. Notice the fact that our shallow neural network function approximation is already more expressive than linear MDPs [5] and thus subsumes tabular MDPs by choosing the features as one-hot representations. Furthermore, realizability is a common assumption that is adopted in almost all theoretic study of RL with function approximation [4,5,6].
>
> 2. "Could the authors clarify on the dependence on H, more explicitly if possible?"
>
>     - Notice that H only influences the variance $\vartheta$ of noises in neural network samples (Eq.(2)). From the proof of concentration results Lemma A.16, Lemma A.17 and Lemma A.18 we know that the sample complexity dependence on variance is $\vartheta^p$, where $p$ is the degree of homogeneous polynomials that can bound the derivative of activation function (defined in Assumption A.4). Therefore the dependence on H is $H^{2p}$​​​.
>
> 3. "Theorem 5.2 states that there is a unique solution to the system of equations. Is there an efficient algorithm for it?"
>
>     - We are not aware of efficient algorithms to solve this recovery problem. Thank you for raising this interesting future question. We will clarify in the Theorem that this is sample efficient, but not computationally efficient.
>
> 4. "A large portion of the results in the neural network recovery is from Zhong et al. 2017, as a clarification is assumption A11 satisfied by ReLu activation?"
>
>     - Yes, it is shown in [Zhong et al. 2017] that ReLU, leaky ReLU and squared ReLU all satisfy this assumption.
>
>
>
> [1] Levine, S., C. Finn, T. Darrell, and P. Abbeel. 2016. “End-to-end training of deep visuomotor policies”. Journal of Machine Learning Research. 17(39): 1–40
>
> [2] You, Y., X. Pan, Z. Wang, and C. Lu. 2017. “Virtual to Real Reinforcement Learning for Autonomous Driving”. arXiv preprint arXiv:1704.03952.
>
> [3] Gauci, J., E. Conti, Y. Liang, K. Virochsiri, Y. He, Z. Kaden, V. Narayanan, and X. Ye. 2018. “Horizon: Facebook’s Open Source Applied Reinforcement Learning Platform”. arXiv preprint arXiv:1811.00260.
>
> [4] Wang, R., Salakhutdinov, R., and Yang, L. F. Provably efficient reinforcement learning with general value function approximation. arXiv preprint arXiv:2005.10804. 2020.
>
> [5] Jin, C., Yang, Z., Wang, Z., and Jordan, M. I. Provably efficient reinforcement learning with linear function approximation. In Conference on Learning Theory, pp. 2137–2143. PMLR, 2020.
>
> [6] Jin, C., Liu, Q., and Miryoosefi, S. Bellman eluder dimension: New rich classes of rl problems, and sample-efficient algorithms. arXiv preprint arXiv:2102.00875, 2021.

---

> > ### Comment · Reviewer_qeUq · 2021-08-23
> > **Thank you for the clarifications**
> >
> > Dear authors,
> >
> >  Thank you for the response which has successfully clarified many points. After carefully reading the other reviews and the detailed responses I have an obligation to raise my scores as the rebuttal was solid in explaining almost all of the concerns of the reviewers.
> >
> > One small clarification regarding the dense features assumption:
> >
> > I see that this is essential for good exploration and the also in the NN parameter recovery. My questions are more of a practical nature:
> >
> > 1) We obtain n iid samples from the standard Gaussian in the d-dimensions and then query the corresponding (s,a)-state action pair. Now do we have an invertible feature mapping? For example, let's say my state space is discrete, but has exponentially many (s,a) pairs.
> > 2)A related question, is there anyway to ensure that we could have a dense feature map in practice?
> >
> > My apologies if I may be missing something trivial.
> >
> > Finally, I think it would be great if the authors could have a short discussion about computational and statistical trade-offs in their algorithms, as this would be tremendously useful to the community.

---

> > > ### Author Response · Authors · 2021-08-25
> > > **Response to the updates.**
> > >
> > > Thank you for acknowledging this paper. Please find the responses to the further questions below:
> > > - The dense feature map appears in the continuous control problem [Lillicrap et al. 2016], which itself is an interesting theoretical problem and is also found in many practical applications such as robotics. In these problems, the action space is often $\mathbb{R}^d$ or a bounded set in $\mathbb{R}^d$ and the action is input directly to the neural function approximations or after some transforms, depending on the task. In these cases, the algorithm can explore with Gaussian features.
> > > - In the neural network model, the algorithm is computationally efficient because the method of moments only needs $\mathrm{poly}(d)$ time. In the low-rank polynomial model, known efficient methods such as UCB, Abe-Long sampling, zeroth-order optimization, and information directed sampling all fail to obtain $O(d)$ upper bounds, even in bandits problems. Other methods for finite arm bandits such as phased elimination takes exponential time in $d$ since it computes the optimal design and rewards in an exponential action set. Thus we conjecture that there exists a computation barrier to achieve $O(d)$ upper bounds.
> > >
> > > [Lillicrap et al. 2016] Continuous control with deep reinforcement learning. Timothy P. Lillicrap et al. 2016.

---

### Decision · Program_Chairs · 2021-09-27

**Decision:**

Accept (Poster)

**Comment:**

This paper has received a lot of discussion.
In the end, the reviewers agree that the paper contributes to the literature and should be accepted.
There are a number of useful comments that the authors should take into account when preparing the camera ready version of the paper.